# Video-BCI: Bayesian Cognitive Integration of Self-Prior Hypotheses for Video Understanding

Xing Xi [1][2]   Peixian Chen [2]   Yu Qiu [1]   Ronghua Luo [1]   Peilin Tong [2]   Jiahao Liang [1]

## Abstract

Recent progress in vision-language models (VLMs) has driven significant advances in video understanding. However, existing methods often act as naive empiricists, mapping video input directly to output without any mechanism to introspect or challenge inherent bias. In this work, we challenge this paradigm by reframing video reasoning as a Bayesian cognitive process. We propose Video-BCI (Bayesian Cognitive Integration of Self-Prior Hypotheses), a novel framework that first samples a set of Self-Prior Hypotheses to represent the model's intuitive yet potentially biased cognitive state, and then guides the VLMs to perform a critical integration of these priors. This process encourages the model to challenge erroneous majority consensus in cases of high information divergence and to distill superior reasoning chains from its own prior space. The integration is driven by a composite Cognitive Utility Function comprising two intrinsic learning signals: Dialectical Uncertainty Signal (DUS) and Process Tracing Signal (PTS). The DUS incentivizes correct, non-majority judgments by quantifying both the conflict (entropy) among priors and their consensus-challenging strength. The PTS guides the model to trace and learn from reasoning paths within its own priors that lead to better answers, enabling self-driven procedural knowledge distillation. Extensive experiments on six mainstream benchmarks show that Video-BCI achieves new state-of-the-art (SOTA) results across the board. For example, it surpasses the previous best on the MMVU benchmark by 3.8%.

[1]School of Computer Science & Engineering, South China University of Technology, Guangzhou, China [2]Ant Group, Hangzhou, China. Correspondence to: Jiahao Liang <csjiahliang6@mail.scut.edu.cn>.

*Proceedings of the 43rd International Conference on Machine Learning*, Seoul, South Korea. PMLR 306, 2026. Copyright 2026 by the author(s).

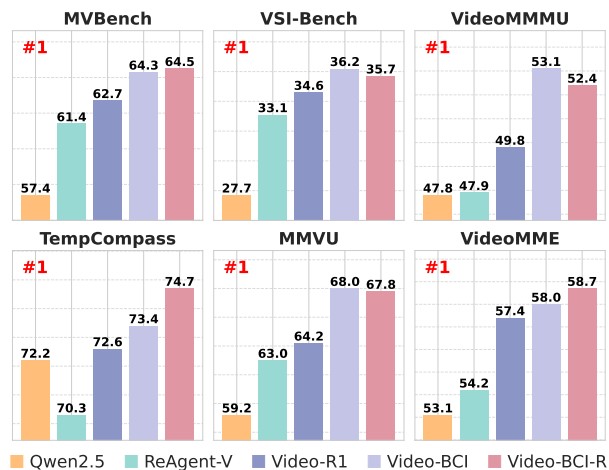

*Figure 1.* **Performance comparison on video understanding benchmarks (16 frames).** Qwen2.5 denotes the multimodal models from the Qwen 2.5 series. Video-BCI and Video-BCI-R represent our non-inference and inference versions, respectively. See the experiments section for further performance analysis.

## 1. Introduction

Video understanding, as a core task in computer vision, is widely applied in key scenarios such as human–computer interaction (Natalie et al., 2025; Guo et al., 2025), autonomous driving (Boroujeni et al., 2025; Wu & Luo, 2025), and intelligent surveillance (Benschop et al., 2025; Yang et al., 2025b). Early approaches typically treated videos as sequences of continuous frames, integrating multi-frame information through temporal attention mechanisms to capture dynamic semantics (Arnab et al., 2021; Sun et al., 2019). In recent years, the rapid advancement of large language models (LLMs) has substantially accelerated progress in this field. Researchers have employed supervised fine-tuning (SFT) techniques to successfully map visual features into the language space, thereby granting LLMs enhanced capabilities in visual comprehension (Zhao et al., 2023; 2024). Meanwhile, other studies have leveraged the strong reasoning ability of LLMs to transform visual inputs into textual descriptions, enabling the construction of text-based agents for complex video-related tasks (Wang et al., 2024; Zhang et al., 2024a). More recently, reinforcement learning (RL)

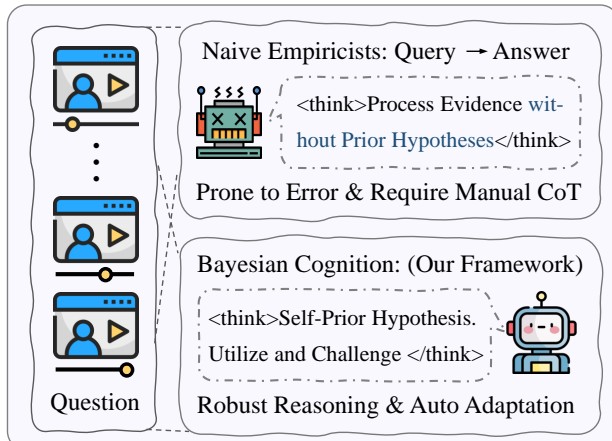

*Figure 2.* **Comparison of the Naive Empiricist (Top) and our Bayesian Cognition (Bottom).** (Top) Naive models process evidence without prior hypotheses, which is error-prone and needs CoT. (Bottom) Our framework uses self-prior hypothesis and learns to use or challenge it, for robust reasoning and adaptation.

has emerged as a mainstream method to further improve performance of vision–language models (VLMs).

To address the vulnerability of VLMs in complex reasoning tasks, recent studies have attempted to design datasets that explicitly expose and rectify model deficiencies. For example, Video-R1 (Feng et al., 2025) disrupts the temporal order of video frames, forcing the model to learn to recover the original sequence structure. VersaVid-R1 (Chen et al., 2025a) inserts black-screen segments into the video and requires the model to infer the missing content, thereby enhancing its ability to model temporal causality. While these methods achieve certain improvements, they operate under a shared paradigm: the assumption that robust reasoning capability can be acquired by passively responding to external evidence. They treat VLMs as naive empiricists that directly map video to answer without constructing or leveraging prior hypotheses (Figure 2 (Top)). This paradigm results in various deficiencies when models are confronted with complex reasoning scenarios. More critically, overcoming these deficiencies often depends on large-scale collections of manually crafted adversarial or diagnostic samples, which are costly to build.

In this paper, we challenge the above paradigm and reconstruct video reasoning as a Bayesian cognitive process. To this end, we propose the Video-BCI (Figure 2 (Bottom)). Instead of passively responding to external evidence, our approach trains VLMs to first generate and then critically integrate their own internal beliefs. The design of this framework is inspired by Bayesian cognitive science, which models human understanding as a continual process of updating prior beliefs in light of new evidence. Following this intuition, Video-BCI operates through a two-stage, self-reflective mechanism. First, we sample a set of prior hypotheses, which serve as an explicit representation of the model's intuitive—yet potentially biased—cognitive state. Second, rather than acting as naive empiricists, VLMs are trained to become critical integrators these priors as internal evidence together with external video content in a dialectical manner. To effectively guide this critical integration process, we design a composite Cognitive Utility Function (CUF) comprising two key intrinsic learning signals: Dialectical Uncertainty Signal (DUS) and Process Tracing Signal (PTS). DUS encourages the model to challenge erroneous majority consensus, particularly in scenarios where its own priors exhibit strong internal conflict (i.e., high entropy). It quantifies both the degree of conflict among prior hypotheses and the strength of disagreement from mainstream consensus. PTS guides the model to trace and learn superior reasoning trajectories embedded within its own hypothesis space that lead to more accurate answers. This enables fully self-driven process knowledge distillation.

To evaluate the effectiveness of the Video-BCI, we performed extensive experiments on six major video understanding benchmarks, covering a range of tasks from general scene understanding to complex causal reasoning. The results clearly indicate that our method secures significant and consistent performance advantages. For instance, on VideoMMMU, Video-BCI improves the model accuracy from 49.8 to 53.1, yielding an absolute gain of 3.3%, and surpasses all previous state-of-the-art (SOTA) methods.

Our primary contributions are summarized below:

- We reformulate video understanding as a Bayesian cognitive process, enabling models to integrate self-prior hypotheses and actively challenge intuitive biases.

- We propose the Video-BCI, which samples self-prior hypotheses and guides VLMs through dialectical integration, eliminating the need for CoT annotations.

- We design a guidance objective with two signals: DUS for challenging incorrect majority views and PTS for refining and transferring high-quality reasoning paths.

- To evaluate the effectiveness of the Video-BCI, we conduct extensive evaluations on six video understanding benchmarks. Video-BCI achieves significant improvements and sets new SOTA, including a 3.8% absolute gain on MMVU over the previous best method.

## 2. Related Works

### 2.1. Video Understanding

Video Understanding (VU) aims to enable machines to parse dynamic events and causal relationships from pixel-level inputs. Early methods extended image understanding models

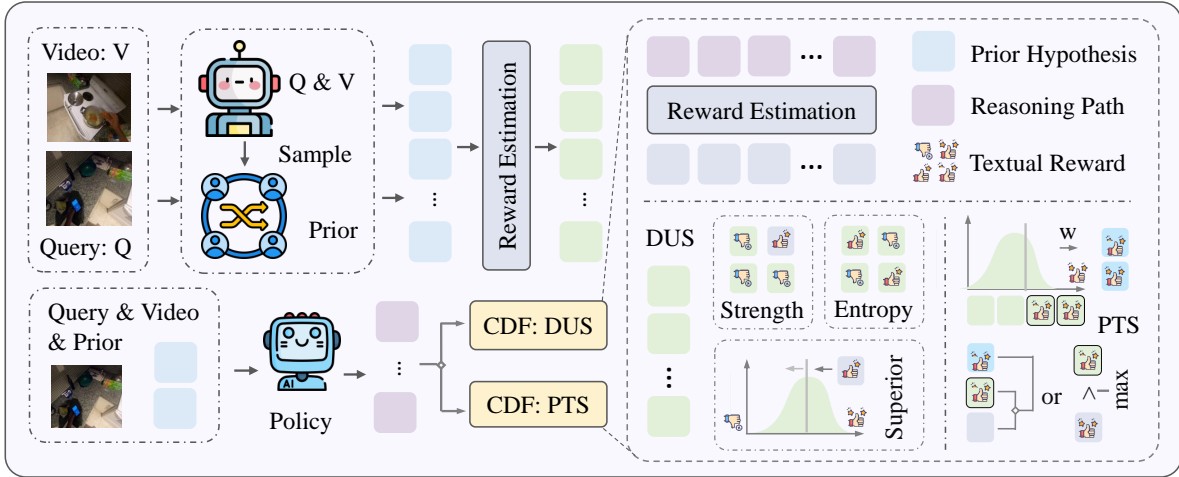

*Figure 3.* **An overview of our Video-BCI framework.** A base sampler model first generates Prior Hypotheses (blue) from the Video (V) and Query (Q). The Policy model acts as an integrator, combining V, Q, and these priors to output final Reasoning Paths (purple). In parallel, Textual Rewards (green) are also computed from the hypotheses via Reward Estimation. Finally, the Cognitive Utility Function (CDF) calculates our intrinsic DUS and PTS signals (detailed right), which in turn train the Policy model for robust reasoning.

by introducing temporal modeling modules to handle the time dimension (Xie et al., 2018; Zolfaghari et al., 2018; Diba et al., 2017; Zhang et al., 2020; Feichtenhofer, 2020). The rise of LLMs has brought a fundamental shift to this field, giving rise to two mainstream paradigms. The first paradigm builds end-to-end VLMs. These methods project visual features into the word embedding space of LLMs by incorporating visual encoders and mapping modules, allowing the models to directly process multimodal inputs and generate textual responses (Zhao et al., 2023; Lin et al., 2023; Zhu et al., 2023). The second paradigm develops LLM-based agents, which decouple perception and reasoning. They first use vision models to convert video content into textual descriptions and then feed these descriptions into pure-text LLMs for subsequent reasoning and decision-making (Wang et al., 2025b; Ataallah et al., 2024b;a). While agent-based approaches can fully exploit the strong text-based reasoning ability of LLMs, they may lose critical visual details during the perception stage. Our work, Video-BCI, follows the first paradigm. On top of end-to-end VLMs, we introduce a post-training stage designed to further enhance their reasoning capabilities.

### 2.2. Reinforcement Learning of VLMs

Recently, inspired by the success of DeepSeek-R1 (DeepSeek-AI, 2025) in reinforcement learning, many studies have begun to explore the transfer of RL strategies to video understanding tasks, with a particular focus on improving VLMs performance through optimization methods such as GRPO. Video-R1 (Feng et al., 2025) disrupts the order of input video frames, forcing the model to infer and recover the original temporal structure to enhance causal

relationship modeling. VersaVid-R1 (Chen et al., 2025a) inserts blank frames to mask parts of the video and requires the model to predict and complete the missing content, thereby improving its ability to hypothesize about unseen events. GRPO-CARE (Chen et al., 2025b) optimizes based on output confidence, guiding the model to produce more certain answers. Several approaches also build VLMs as tool-use agents, leveraging external resources to assist in complex reasoning tasks. Although these methods improve performance in specific aspects, they remain within a naive empiricist paradigm: the model learns only a direct mapping from input to output, without actively questioning or revising its own reasoning. However, Video-BCI leverages a Bayesian cognitive process to make the model generate self-prior hypotheses before reasoning and then critically integrate these priors with external evidence. This process reduces dependence on one-way external data and enables greater generalization in complex reasoning tasks.

## 3. Method

As shown in Figure 3, our Video-BCI framework implements our Bayesian cognitive approach. The process begins when a base sampler model generates a set of Prior Hypotheses (blue blocks) from the external Video (V) and Query (Q). In parallel, a Reward Estimation module computes Textual Rewards (green blocks) associated with these hypotheses. The core of our framework is the Policy model, which acts as a critical integrator. It is trained to synthesize the external (V, Q) with these internal Priors to produce the final Reasoning Paths (purple blocks). This Policy is trained using our Cognitive Utility Function (CUF). As detailed in the zoom-in panel, the CUF provides intrinsic learning signals, the

Dialectical Uncertainty Signal (DUS) and the Process Tracing Signal (PTS). These signals evaluate the model's own outputs to guide policy toward robust reasoning, without human-annotated steps.

---

**Algorithm 1** Dialectical Uncertainty Signal (DUS)

---

0: **Input:**
    Current answer $a$,
    Prior answer set $\mathcal{H}_a$,
    Pre-computed accuracy score $acc_a > 0$,
    Pre-computed prior scores $\{acc_i\}$ for $a_i \in \mathcal{H}_a$,
    Boolean flag $is\_discrete$.
0: **Output:** DUS.
0: **if** $is\_discrete$ **then**
0:     **Step 1: Compute Cognitive Conflict $D$.**
0:     $H \leftarrow -\sum_{a_j \in \mathcal{H}_a^{\text{unique}}} P(a_j) \log_2(P(a_j))$. {Eq. (6)}
0:     $D \leftarrow H / \log_2(|\mathcal{H}_a^{\text{unique}}|)$. {Eq. (7)}
0:     **Step 2: Compute Challenge Strength $S$.**
0:     $N \leftarrow |\mathcal{H}_a|$.
0:     $P_a \leftarrow \text{Frequency}(a, \mathcal{H}_a)/N$.
0:     $S \leftarrow 1 - P_a$. {Eq. (8)}
0:     DUS $\leftarrow D \cdot S$. {Eq. (5)}
0: **else**
0:     **Step 1: Compute Relative Superiority.**
0:     $better\_count \leftarrow 0$.
0:     **for** $acc_i$ in $\{acc_i\}$ **do**
0:         **if** $acc_a > acc_i$ **then**
0:             $better\_count \leftarrow better\_count + 1$.
0:         **end if**
0:     **end for**
0:     DUS $\leftarrow better\_count/|\mathcal{H}_a|$. {Eq. (9)}
0: **end if**
0: **return** DUS. =0

---

### 3.1. Bayesian Cognitive Process

We model the video understanding task as a reasoning process within a structured hypothesis space, performed by a Bayesian cognitive agent. The cognitive state of this agent is characterized by a joint prior measure:

$$\mu_0 \in \mathcal{P}(\mathcal{A} \times \mathcal{R} \mid \mathcal{V} \times \mathcal{Q}), \qquad (1)$$

Here, $\mathcal{V}$, $\mathcal{Q}$, $\mathcal{A}$, and $\mathcal{R}$ represent the measurable spaces for the video, question, answer, and reasoning chain, respectively, and $\mathcal{P}$ denotes the probability measure space. This prior encodes the agent's initial belief about which answer-reasoning pairs are plausible in the given visual-linguistic context. Since the analytical form of $\mu_0$ is unavailable, we draw a set of self-prior hypotheses by sampling from a parameterized base model $\pi_0$ (i.e., pre-trained Qwen 2.5 VL):

$$\mathcal{H} = \left\{ (a^{(i)}, r^{(i)}) \right\}_{i=1}^{K} \sim \pi_0(a, r \mid v, q), \qquad (2)$$

This set of samples $\mathcal{H}$ is a discrete empirical approximation of $\mu_0$. Despite $\pi_0$'s strong multi-modal representation capability, the induced prior $\mu_0$ may contain systematic biases. In $\mathcal{H}$, such biases often appear as a concentrated majority consensus, high-frequency hypotheses with large prior probabilities that contradict the true causal mechanism. Given observed evidence $(v, q, a^*)$ ($a^*$ as the ground-truth answer), an ideal cognitive agent should infer a posterior policy $\pi^*$ that aligns with the evidence and corrects these structural bias errors. Formally, this rule is defined as:

$$\pi^* = \arg\max_{\pi \in \Pi} \; \mathbb{E}_{(a,r) \sim \pi(\cdot|v,q,\mathcal{H})} \left[ \mathcal{U}(a, r; a^*, \mathcal{H}) \right]. \quad (3)$$

Here, $\Pi$ is the set of all computable policies. $\mathcal{U}(\cdot)$ is the cognitive guidance function, which we define as a composite function combining a standard reinforcement learning reward with our proposed cognitive signals:

$$\mathcal{U}(a, r; a^*, \mathcal{H}) = A(a, a^*) + F(a) + \text{CUF}(a, r; a^*, \mathcal{H}), \quad (4)$$

where $A(\cdot)$ and $F(\cdot)$ measures accuracy and format, respectively. In practice, we parameterize this policy as a trainable visual-language model $\pi_\theta(a, r \mid v, q, \mathcal{H})$. Its input explicitly includes the video $v$, the question $q$, and the complete set of hypotheses $\mathcal{H}$. This design allows the model to directly access the statistical structure of the prior hypotheses, including the central tendency of the answer distribution, the degree of group disagreement (measured by entropy), and the semantic diversity of the reasoning chains.

### 3.2. Cognitive Utility Function

Cognitive Utility Function converts the Bayesian integration process into learnable signals that drive the model to critically reflect on its self-prior hypotheses. Unlike traditional objectives that rely only on answer correctness, the CUF has two components: DUS: This rewards the model for correctly deviating from the incorrect majority consensus, especially when the prior hypothesis group shows high disagreement. PTS: This guides the model to learn superior reasoning paths from its own priors, enabling knowledge distillation. This design encourages the model to challenge incorrect consensus and transforms passive prediction into active reflection. The pseudocode for DUS and PTS is presented in Algorithm 1 and Algorithm 2, respectively.

**Dialectical Uncertainty Signal (DUS).** The core objective of the DUS is to equip the VLMs with critical thinking. Through DUS, we guide the model to: identify cognitive conflict by sensing high information entropy in the self-prior set $\mathcal{H}$; and challenge incorrect consensus by proposing correct minority opinions when the prior beliefs form a wrong majority. To prevent reward hacking (e.g., outputting random answers simply to gain rewards), DUS is activated only when the model's final answer is verified as correct.

Formally, for a given posterior judgment $(a, r)$, the DUS is:

$$\text{DUS}(a; \mathcal{H}, a^*) = \mathbb{I}(A(a, a^*) > 0) \cdot D(\mathcal{H}_a) \cdot S(a, \mathcal{H}_a), \quad (5)$$

where $\mathbb{I}(\cdot)$ is an indicator function, ensuring this term is computed only when the base accuracy is positive. $D(\mathcal{H}_a)$ and $S(a, \mathcal{H}_a)$ are our two core metrics: Cognitive Conflict and Consensus Challenge Strength, respectively. $\mathcal{H}_a = \{a^{(1)}, \ldots, a^{(K)}\}$ is the set of answers extracted from the prior hypothesis set $\mathcal{H}$. We use normalized entropy to quantify $D(\mathcal{H}_a)$ in the prior hypothesis $\mathcal{H}_a$. A high-entropy set signals a difficult problem that requires deliberate reasoning. We first calculate the $H(\mathcal{H}_a)$ as:

$$H(\mathcal{H}_a) = -\sum_{a_j \in \mathcal{H}_a^{\text{unique}}} P(a_j) \log_2(P(a_j)), \quad (6)$$

Here, $\mathcal{H}_a^{\text{unique}}$ is the set of unique answers in $\mathcal{H}_a$, and $P(a_j)$ is the empirical probability of answer $a_j$ in $\mathcal{H}_a$, which is approximated by its frequency in the set. We then define the final Cognitive Conflict $D(\mathcal{H}_a)$ as the ratio of this entropy to its theoretical maximum. This normalizes its value to the range $[0, 1]$ for cross-problem comparability:

$$D(\mathcal{H}_a) = H(\mathcal{H}_a) / \log_2(|\mathcal{H}_a^{\text{unique}}|). \quad (7)$$

The Consensus Challenge Strength $S(a, \mathcal{H}_a)$ measures how much the posterior answer $a$ challenges the mainstream consensus in the prior beliefs. We define this as the complement of the empirical probability of answer $a$ in the hypothesis set. A high strength value means the agent identified a correct hypothesis that had a low prior probability.

$$S(a, \mathcal{H}_a) = 1 - P(a) = 1 - N(a) / |\mathcal{H}_a|. \quad (8)$$

Here, $N(a)$ is the count of answer $a$ in $\mathcal{H}_a$, and $|\mathcal{H}_a|$ is the total number of prior answers. If the verified correct answer $a$ is a minority opinion, its challenge strength $S(a, \mathcal{H}_a)$ will approach 1. Conversely, if it follows the majority opinion, its strength will approach 0. However, the implementation above (based on $P(a)$) is only usable in classification scenarios. It fails when extended to non-classification settings, such as open-ended generation. To apply DUS in more scenarios, we use a relative incremental percentage for non-classification problems. This is the proportion of answers in the prior set that the current answer $a$ is better than, which encourages the VLM to challenge the prior:

$$\text{DUS}(a; \mathcal{H}, a^*) = \frac{1}{|\mathcal{H}_a|} \sum_{a^{(i)} \in \mathcal{H}_a} \mathbb{I}(\bar{A}(a) > \bar{A}(a^{(i)})). \quad (9)$$

Here, $\bar{A}(\cdot)$ represents the accuracy score for non-multiple-choice tasks (e.g., ROUGE-L). This calculates the percentage of prior answers that current answer $a$ is superior to.

**Process Tracing Signal (PTS).** PTS is to achieve a form of self-driven process knowledge distillation. Unlike DUS, which focuses on the correctne ss and challenge of the answer $a$, PTS focuses on the quality of the reasoning path $r$ (i.e., the think content in the code). The core idea is to guide the policy model $\pi_\theta$'s reasoning path $r$ to trace and learn from the superior reasoning paths $r^{(i)}$ that are present in the prior hypothesis set $\mathcal{H}$ and lead to better answers $a^{(i)}$. Specifically, we first define a superior hypothesis set $\mathcal{H}^+$. This set contains all hypotheses from the prior $\mathcal{H}$ whose accuracy is higher than the accuracy of the current answer:

$$\mathcal{H}^+ = \{(a^{(i)}, r^{(i)}) \in \mathcal{H} \mid \bar{A}(a^{(i)}) > \bar{A}_{acc}(a)\}. \quad (10)$$

For classification problems, we use $A(\cdot)$ to calculate the answer accuracy. $U_{PTS}$ is defined as the weighted average of the similarity between the current reasoning $r$ and all reasoning paths $r^{(i)}$ in $\mathcal{P}^+$. This design encourages the $r$ generated by $\pi_\theta$ to imitate the $r^{(i)}$ that have been shown (in the prior) to yield higher accuracy. We introduce a key mechanism: the weight $w^{(i)}$. This weight is proportional to the "accuracy improvement" of $a^{(i)}$ relative to $a$. This means the model is more strongly encouraged to imitate reasoning paths that lead to the largest performance gains:

$$w^{(i)} = (\bar{A}(a^{(i)}) - \bar{A}(a)) \cdot \max(\epsilon, \bar{A}_\Delta - \bar{A}(a))^{-1}. \quad (11)$$

Here, $A_\Delta$ is the highest accuracy among all hypotheses in $\mathcal{H}$. $\epsilon$ is a small value, $10^{-5}$, to prevent division by zero. The final PTS reward is calculated using normalized weights:

$$\text{PTS}(a; \mathcal{H}, a^*) = \sum_{(a^{(i)}, r^{(i)}) \in \mathcal{H}^+} \tilde{w}^{(i)} \cdot \text{Sim}(r, r^{(i)}), \quad (12)$$

where $\tilde{w}^{(i)} = w^{(i)} / \sum_{j \in \mathcal{H}^+} w^{(j)}$. $\text{Sim}(r, r^{(i)})$ is the similarity score between the current reasoning $r$ and the prior reasoning $r^{(i)}$. Special Case: If $\mathcal{H}^+$ is empty, it means the current answer $a$ is already the best (or tied for the best) among all prior hypotheses. In this situation, the model directly receives the maximum reward to encourage it to maintain this optimal reasoning state.

## 4. Experiment

### 4.1. Setup

**Benchmark and Metric.** We align with previous work (Feng et al., 2025; Chen et al., 2025b) and use Video-R1-260k as our training set. Notably, we do not perform SFT before reinforcement learning. Thus, we only use prompt and answer portions of this dataset. We evaluate Video-BCI on six benchmarks, which fall into two categories. Reasoning Benchmarks: VSI-Bench (Yang et al., 2025a), VideoM-MMU (V-MMMU)(Hu et al., 2025), and MMVU (Zhao et al., 2025). General-Purpose Benchmarks: MVBench (Li et al., 2024a), TempCompass (TC) (Liu et al., 2024b), and VideoMME (Fu et al., 2025). Following prior work, we only evaluate multiple-choice subset of MMVU. For VideoMME,

*Table 1.* **Performance comparison with recent SOTA methods on video reasoning and general-purpose benchmarks.** Proprietary denotes closed-source commercial models. SFT, LLM Agent, and RL indicate the method types (Supervised Fine-Tuning, LLM-based Agent, and Reinforcement Learning, respectively). Video-BCI-R-7B is our full reasoning-enhanced model, while Video-BCI-7B is our baseline variant (without the R component). All models (RL) are evaluated at a 256×28×28 resolution for fair comparison.

| Method | Pub | Frame | Video Reasoning Benchmark | | | Video General Benchmark | | |
| --- | --- | --- | --- | --- | --- | --- | --- | --- |
| | | | VSI-Bench | V-MMMU | MMVU | MVBench | TC | VideoMME |
| ○ Proprietary Models | | | | | | | | |
| GPT-4o | - | - | 34.0 | 61.2 | 75.4 | - | - | 71.9 |
| Gemini 1.5 Pro | - | - | - | 53.4 | 71.2 | - | - | 75.0 |
| ○ SFT-Based Models & LLM Agent | | | | | | | | |
| LLaMA-VID (Li et al., 2024b) | ECCV 24 | - | - | - | - | 41.9 | 45.6 | - |
| VideoLLaMA2 (Zhu et al., 2023) | arXiv 24 | - | - | - | 44.8 | 54.6 | - | 47.9 |
| LongVA-7B (Zhang et al., 2024b) | arXiv 24 | - | 29.2 | 23.9 | - | - | 56.9 | 52.6 |
| LLaVA-OneVision (Li et al., 2025) | PMLR 24 | - | 32.4 | 33.8 | 49.2 | 56.7 | - | 58.2 |
| Kangaroo-8B (Liu et al., 2024a) | Arxiv 24 | - | - | - | - | 61.1 | 62.5 | 56.0 |
| ShareGPT4Video (Chen et al., 2024) | NeurIPS 24 | - | - | - | - | 51.2 | - | 39.9 |
| VILA-1.5-8B (Lin et al., 2024) | CVPR 24 | - | 28.9 | 20.8 | - | - | 58.8 | - |
| VILA-1.5-40B (Lin et al., 2024) | CVPR 24 | - | 31.2 | 34.0 | - | - | - | 60.1 |
| Video-UTR-7B (Yu et al., 2025) | ICLR 25 | - | - | - | - | 58.8 | 59.7 | 52.6 |
| VideoTree (GPT 4) (Wang et al., 2025b) | CVPR 25 | 64 | - | 47.8 | 54.2 | - | - | 56.1 |
| mPLUG-Owl3-8B (Ye et al., 2025) | ICLR 25 | - | - | - | - | 54.5 | - | 53.5 |
| ○ RL-Based Models | | | | | | | | |
| RRPO-7B (Sarkar & Etemad, 2025) | NeruIPS 25 | 16 | - | - | - | 59.0 | 44.3 | - |
| ReAgent-V (Zhou et al., 2025) | NeruIPS 25 | 16 | 33.1 | 47.9 | 63.0 | 61.4 | 70.3 | 54.2 |
| Video-R1-7B (Feng et al., 2025) | NeruIPS 25 | 16 | 34.6 | 49.8 | 64.2 | 62.7 | 72.6 | 57.4 |
| **Ours:** Video-BCI-7B | - | 16 | **36.2** | **53.1** | **68.0** | 64.3 | 73.4 | 58.0 |
| **Ours:** Video-BCI-R-7B | - | 16 | 35.7 | 52.4 | 67.8 | **64.5** | **74.7** | **58.7** |
| Video-R1-7B (Feng et al., 2025) | NeruIPS 25 | 32 | 35.8 | 52.3 | 63.8 | 63.9 | 73.2 | 59.3 |
| GRPO-CARE-7B (Chen et al., 2025b) | Arxiv 25 | 32 | 35.8 | 50.4 | 65.8 | 65.1 | 73.5 | 59.6 |
| **Ours:** Video-BCI-7B | - | 32 | **38.2** | 53.2 | **68.0** | 66.1 | **74.2** | 60.2 |
| **Ours:** Video-BCI-R-7B | - | 32 | 36.7 | **54.2** | **68.0** | **66.3** | 73.5 | **60.5** |
| Video-R1-7B (Feng et al., 2025) | NeruIPS 25 | 64 | 37.1 | 52.4 | 63.8 | 64.8 | 73.2 | 61.4 |
| Video-RTS-7B (Wang et al., 2025a) | EMNLP 25 | 64 | - | 52.7 | 66.3 | - | - | - |
| **Ours:** Video-BCI-7B | - | 64 | **39.9** | 53.0 | 67.8 | **68.0** | 74.2 | 62.6 |
| **Ours:** Video-BCI-R-7B | - | 64 | 39.4 | **54.4** | **69.6** | 67.8 | 73.8 | **62.9** |

we do not use its captions, which allows us to assess pure visual understanding capabilities. For metrics, we align with accuracy metrics used during our training phase.

**Implementation Details.** We conduct our experiments on 16 A100 GPUs (Total 1280 G). We use the instruction-finetuned version of Qwen2.5-VL (Bai et al., 2025) as our base model. Following prior work, we set the video resolution to 128×28×28 during training and uniformly sample 16 frames. The training learning rate is set to 1e-6, with a total batch size of 16 (a per-device batch size of 1). The number of prior samples and the number of rollouts are both set to 8. During inference, the resolution is set to 256×28×28 to ensure a fair comparison. For our DUS implementation, we use combined cognitive conflict and challenge strength

for multiple-choice problems but apply relative incremental percentage for others.

### 4.2. Main Results

We evaluate our Video-BCI framework on six benchmarks, as shown in Table 1. Our method is assessed under two main variants: Video-BCI-7B (our base cognitive model with DUS) and Video-BCI-R-7B (our full reasoning model).

**Comparison with Proprietary Models.** Our Video-BCI-R-7B (64-frame) shows highly competitive performance against proprietary models. It approaches top-tier models like GPT-4o and Gemini 1.5 Pro on general benchmarks (e.g., VideoMME) and even surpasses them on key reasoning benchmarks like VSI-Bench (39.4% vs 34.0%) and

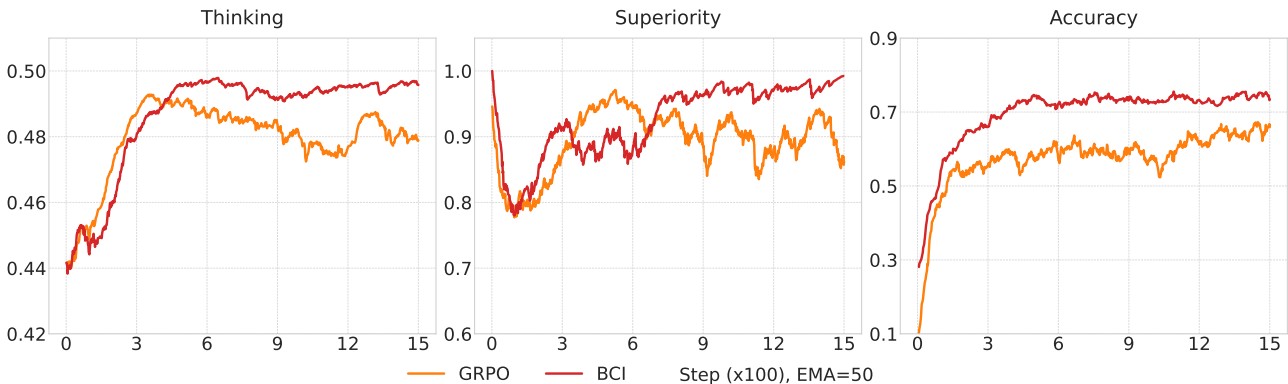

*Figure 4.* **Training process comparison of our Video-BCI (BCI) against the GRPO baseline.** The x-axis represents training steps (x100), and all curves are smoothed with an EMA of 50. The three plots show: (Left) Thinking: This metric is the PTS score, indicating the correlation between the VLM's generated sequence and high-quality priors. (Middle) Superiority: This metric indicates the relative accuracy ranking (percentile) of the model's generated answers within the prior answer set. (Right) Accuracy: The accuracy score.

*Table 2.* **Ablation study of the Video-BCI framework.** CoT and MJ are the Chain-of-Thought and Majority Voting baselines, respectively. BCI is a basic variant of our Bayesian process. BCI-GRPO is the standard GRPO baseline without DUS or PTS. BCI-DUS and BCI-PTS represent adding only the DUS or PTS, respectively. BCI-R is the complete reasoning model. See the appendix for more experiments.

| Method | Frame | Video Reasoning Benchmark | | | Video General Benchmark | | |
|---|---|---|---|---|---|---|---|
| | | VSI-Bench | VideoMMMU | MMVU | MVBench | TempCompass | VideoMME |
| Qwen 2.5-VL-CoT | 16 | 27.7 | 47.8 | 59.2 | 57.4 | 72.2 | 53.1 |
| Qwen 2.5-BCI | 16 | 28.8 | 48.8 | 66.3 | 61.9 | 72.8 | 55.6 |
| Qwen 2.5-VL-MJ | 16 | 28.5 | 49.9 | 65.0 | 62.9 | 71.6 | 55.0 |
| Qwen 2.5-BCI-GRPO | 16 | 32.0 | 50.3 | 64.8 | 61.2 | 72.3 | 53.5 |
| Qwen 2.5-BCI-DUS | 16 | 36.2 | 53.1 | 68.0 | 64.3 | 73.4 | 58.0 |
| Qwen 2.5-BCI-PTS | 16 | 34.9 | 51.2 | 68.2 | 64.4 | 74.5 | 57.2 |
| Qwen 2.5-BCI-R | 16 | 35.7 | 52.4 | 67.8 | 64.5 | 74.7 | 58.7 |

VideoMMMU (54.4% vs 53.4%). This highlights our framework's strength in complex reasoning.

**Comparison with SFT and Agent Methods.** Our Video-BCI framework demonstrates a significant advantage over methods that rely on SFT or complex LLM Agent architectures. On both reasoning and general benchmarks, our 64-frame model comprehensively outperforms prior SOTA. On MVBench, our Video-BCI-R-7B (67.8%) is significantly higher than Kangaroo-8B (61.1%) and Video-UTR-7B (58.8%). This indicates our RL approach can more effectively stimulate deep reasoning than SFT or Agent.

**Comparison with RL Baselines.** As shown by Table 1, Video-BCI-R-7B comprehensively surpasses the previous best RL baseline (underlined, Video-R1-7B) across all frame settings and benchmarks. This lead is all-encompassing, with significant gains on both reasoning (e.g., +3.3% on MMVU-64) and general tasks (e.g., +3.2% on MVBench-64). Furthermore, comparing our two variants demonstrates the value of PTS. The DUS-only Video-BCI-7B already establishes a strong baseline (e.g., 53.0% on VideoMMMU-

64), but adding PTS to create Video-BCI-R-7B provides a consistent and crucial performance boost (e.g., to 54.4% on VideoMMMU and 62.9% on VideoMME). This demonstrates PTS successfully distills superior reasoning paths.

### 4.3. Training Curves

The training curves in Figure 4 clearly demonstrate the comprehensive advantages of Video-BCI (BCI) over the GRPO baseline. First, on the Thinking metric (left plot), which reflects PTS-guided knowledge distillation, the BCI (red line) converges to a high, stable plateau (around 0.49-0.50). This is significantly higher than the GRPO (orange line) curve, which shows sharp fluctuations and a decline, indicating our PTS signal stably guides the model to learn high-quality reasoning paths. Second, this advantage is reflected in the Superiority metric (middle plot), which measures the quality of the model's answers relative to the priors. After the initial phase, the BCI curve is consistently higher than GRPO's and steadily approaches 1.0. This demonstrates that BCI learns cognitive reflection more quickly, generating responses su-

What is the size of this room (in square meters)? If multiple rooms are shown, estimate the size of the combined space.     Answer: 15

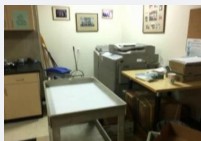 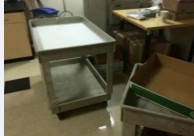 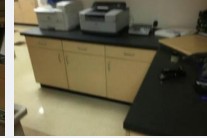 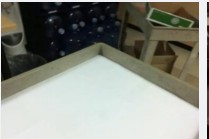 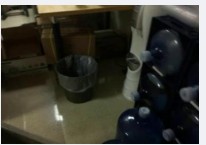 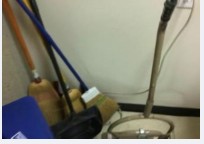

**Qwen25-VL (MJ):** <think>Let me think, combining all visible elements, I feel a single room here is about 12 square meters. For combined areas, we can only estimate one as more information is needed.</think><answer>12</answer>

**Video-BCI:** <think>Solved!</think><answer>15</answer>
**Video-BCI-R**: <think>The analyses provided by the hypotheses all consistently align that the room's size, whether single or combined, falls perfectly within the range of 10 to 15 square meters. This range ensures the room comfortably fits all visible equipment and spaces while leaving room for the slight variations in depth and corner areas not fully captured. This cohesive range perfectly matches estimated size, confirming that room's size, and thus combined space, is ideally 15 square meters. </think><answer>15</answer>

*Figure 5.* **A qualitative case study of Video-BCI on a complex reasoning task.** The Qwen-VL (MJ) baseline (Majority Judgment) fails to handle ambiguity, answering incorrectly (12). Video-BCI (DUS only) challenges the consensus and answers correctly (15). Video-BCI-R (DUS+PTS) reproduces a high-quality reasoning process by analyzing prior hypotheses to find the correct answer.

perior to its initial beliefs. Finally, these dual advantages translate to task performance, as shown in the Accuracy curve (right plot). BCI not only converges faster but also stably converges to an accuracy plateau (around 0.72-0.74), which is higher than GRPO, demonstrating our method's superior performance.

### 4.4. Ablation Study

As shown in Table 2, we conduct a series of ablation studies to verify the effectiveness of our proposed DUS and PTS signals, using BCI-GRPO (standard GRPO training) as the baseline. First, adding either the DUS or PTS signal alone brings significant performance improvements. BCI-DUS (DUS only) shows strong performance across all benchmarks. It brings a large boost of +4.2 ($32.0 \rightarrow 36.2$) on the reasoning benchmark VSI-Bench and an increase of +3.2 on MMVU. This strongly demonstrates the powerful effectiveness of DUS in motivating the model to challenge incorrect consensus. At the same time, BCI-PTS (PTS only) also comprehensively surpasses the baseline. For example, it brings a significant improvement of +3.4 ($64.8 \rightarrow 68.2$) on MMVU and +2.2 on TempCompass. This proves the value of process tracing (learning high-quality think paths). Finally, our full model, BCI-R (using both DUS and PTS), achieves the best overall performance on multiple benchmarks. For example, it outperforms any single component on VideoMME (58.7) and VideoMMMU (52.4).

### 4.5. Case Study

We present a qualitative case study in Figure 5 to illustrate the effectiveness of Video-BCI in complex reasoning. The task requires estimating room size and handling a potential ambiguity. The baseline Qwen25-VL (MJ) (Majority Judg-

ment) performs poorly. Its ¡think¿ process shows it gives up on reasoning (more information is needed) and provides an incorrect answer (12), exposing its vulnerability to ambiguity and incorrect prior consensus. In contrast, our Video-BCI (DUS only) model successfully challenges this consensus and provides the correct answer (15). This demonstrates the DUS signal's effectiveness in guiding the model to select the correct answer. Furthermore, our complete Video-BCI-R (DUS+PTS) model not only answers correctly but also generates a high-quality reasoning path. As shown in its think content, the model explicitly analyzes and integrates the prior hypotheses by establishing a reasonable confidence interval to confirm 15 as the ideal answer. This proves that the PTS signal successfully guides the model to distill and reproduce a superior reasoning process.

## 5. Conclusion

In this paper, we challenge the naive empiricist paradigm in VLM-based video understanding by proposing Video-BCI, a framework that reframes the task as a Bayesian cognitive process. Video-BCI trains a VLM to act as a critical integrator that first samples a set of self-prior hypotheses and then dialectically synthesizes them with external evidence. This process is guided by a composite cognitive utility function with two novel signals: the dialectical uncertainty signal, which incentivizes challenging incorrect majority consensus, and the process tracing signal, which distills superior reasoning paths from the priors. Extensive experiments on six benchmarks show that Video-BCI achieves new SOTA. Ablation studies and qualitative analyses further verify that DUS and PTS are complementary, significantly improving reasoning robustness and transforming the VLM from a passive predictor into an active cognitive reflector.

## Impact Statement

This paper presents work whose goal is to advance the field of Machine Learning. There are many potential societal consequences of our work, none which we feel must be specifically highlighted here.

## Acknowledgements

The authors are supported by the National Key Research and Development Program of China (Grant No. 2024YFE0105400). In addition, this work was supported by Ant Group Research Intern Program. We would like to thank all the anonymous reviewers for their insightful comments.

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

## A. Analysis of Hyperparameter $K$

**Analysis of Video-BCI (DUS-only, Figure 6 Top).** The performance curves for the Video-BCI variant (corresponding to BCI-DUS in the main text) are presented in Figure 6 (top row). These plots reveal the dependency of the DUS on the number of prior hypotheses, $K$. The core mechanism of DUS is to evaluate the cognitive conflict (entropy) and challenge strength (minority correctness) within the answer distribution formed by the $K$ priors. When $K$ is small (e.g., $K = 0$ or $K = 1$), the model lacks sufficient statistical samples to form a meaningful consensus distribution, rendering the DUS signal ineffective. As $K$ increases towards 8, the model obtains more stable statistics, allowing DUS to more accurately quantify conflict and reward the behavior of challenging an incorrect consensus. This effect is particularly evident on the Videommu and MMVU benchmarks, where performance steadily improves with a larger $K$. On other benchmarks, such as Tempcompass, performance saturates around $K \approx 4$. This suggests that 4 priors are sufficient to establish a stable answer distribution for this task, leading to diminishing marginal returns for the DUS signal.

**Analysis of Video-BCI-R (DUS + PTS, Figure 6 Bottom).** The performance curves for Video-BCI-R (our full model), shown in Figure 6 (bottom row), exhibit a distinct and much stronger dependency on $K$. This model incorporates not only DUS but also the PTS, which is designed to actively trace and distill high-quality reasoning paths from the $K$ priors. Consequently, when $K = 0$, the model lacks any reasoning paths to trace. This causes its performance (especially on VSI-Bench and MMVU) to fall even below that of the $K = 0$ DUS-only baseline. However, as $K$ increases, the PTS gains access to richer teaching material in the form of high-quality reasoning paths. This results in steeper performance curves, and the peak performance achieved on most complex reasoning benchmarks (e.g., VSI-Bench) is significantly higher than that of the DUS-only model. This demonstrates that PTS successfully elevates the model's utilization of priors from statistical answer-checking to sophisticated reasoning-path distillation, thereby achieving a higher performance ceiling.

**Summary and Conclusion.** In summary, the ablation study in Figure 6 clearly reveals the dual role of $K$ in our framework and confirms the complementary nature of DUS and PTS. The performance of Video-BCI (DUS-only) improves as $K$ increases because richer answer statistics enhance the judgment of cognitive conflict. In contrast, the performance of Video-BCI-R (DUS+PTS) improves more significantly because it additionally leverages the reasoning paths (i.e., the semantic content) provided by the priors for process distillation. These curves indicate that while our models are uniformly trained with $K = 86$, the optimal $K$ value at inference is task-dependent (e.g., Tempcompass performance

saturates at $K = 4$). The observation that $K = 8$ (or a value close to 8) yields optimal performance on the majority of complex benchmarks validates our training strategy. This approach successfully teaches the model to utilize a rich pool of priors, granting us the flexibility to select the optimal $K$ for each specific benchmark.

## B. Train and Val Template

The prompt template, illustrated in Figure 7, serves as the core mechanism for the Video-BCI framework. It is not merely an input instruction; rather, it is meticulously designed to support Bayesian Cognitive Integration and, crucially, to enable the calculation of DUS and PTS signals.

The template first defines the role of the Policy model, $\pi_\theta$ (as an advanced AI assistant), and includes three key variable placeholders: problem, hypotheses, and format. The hypotheses placeholder is the central technical component of the Video-BCI framework. Our method requires the Policy model, $\pi_\theta$, to receive and reflect upon a set of $K$ prior hypotheses, $\mathcal{H}$. This placeholder functions as the carrier for $\mathcal{H}$. In practice, the $K$ priors sampled from $\pi_0$ (which are wrapped in analysis tags as shown in the Hypotheses Example) are inserted into this placeholder. This provides the Policy model with the internal evidence necessary to perform its critical integration task. Furthermore, the template compels the model to strictly separate its output into two distinct components: (1) the reasoning path $r$, enclosed within think tags, and (2) the final answer $a$, enclosed within answer tags. This structured separation is a critical technical prerequisite. Our PTS operates by computing the similarity between the content of the current think block ($r$) and the superior reasoning paths ($r^{(i)}$) found in the priors. Concurrently, our DUS functions by analyzing the content of the answer block ($a$) for its correctness and its statistical relationship to the prior answer set $\mathcal{H}_a$.

## C. Sampling Template for Prior Hypotheses

Figure 8 presents the prompt template used to instruct the base model ($\pi_0$) in generating the $K$ Self-Prior Hypotheses ($\mathcal{H}$). As indicated by its title, this template is adopted from the Video-R1 (Feng et al., 2025). We utilize this specific prompt as it effectively encourages the model to simulate a human-like, in-depth thought process, often involving internal dialogue (e.g., using phrases such as let me think or let's break it down).Most importantly, this template enforces a structured output, mandating a strict separation between the detailed reasoning path (enclosed within think tags) and the final answer (enclosed within answer tags). This structural requirement is essential for our framework, as it allows us to programmatically extract both the prior answer $a^{(i)}$ and the prior reasoning $r^{(i)}$ from the output of $\pi_0$. Both

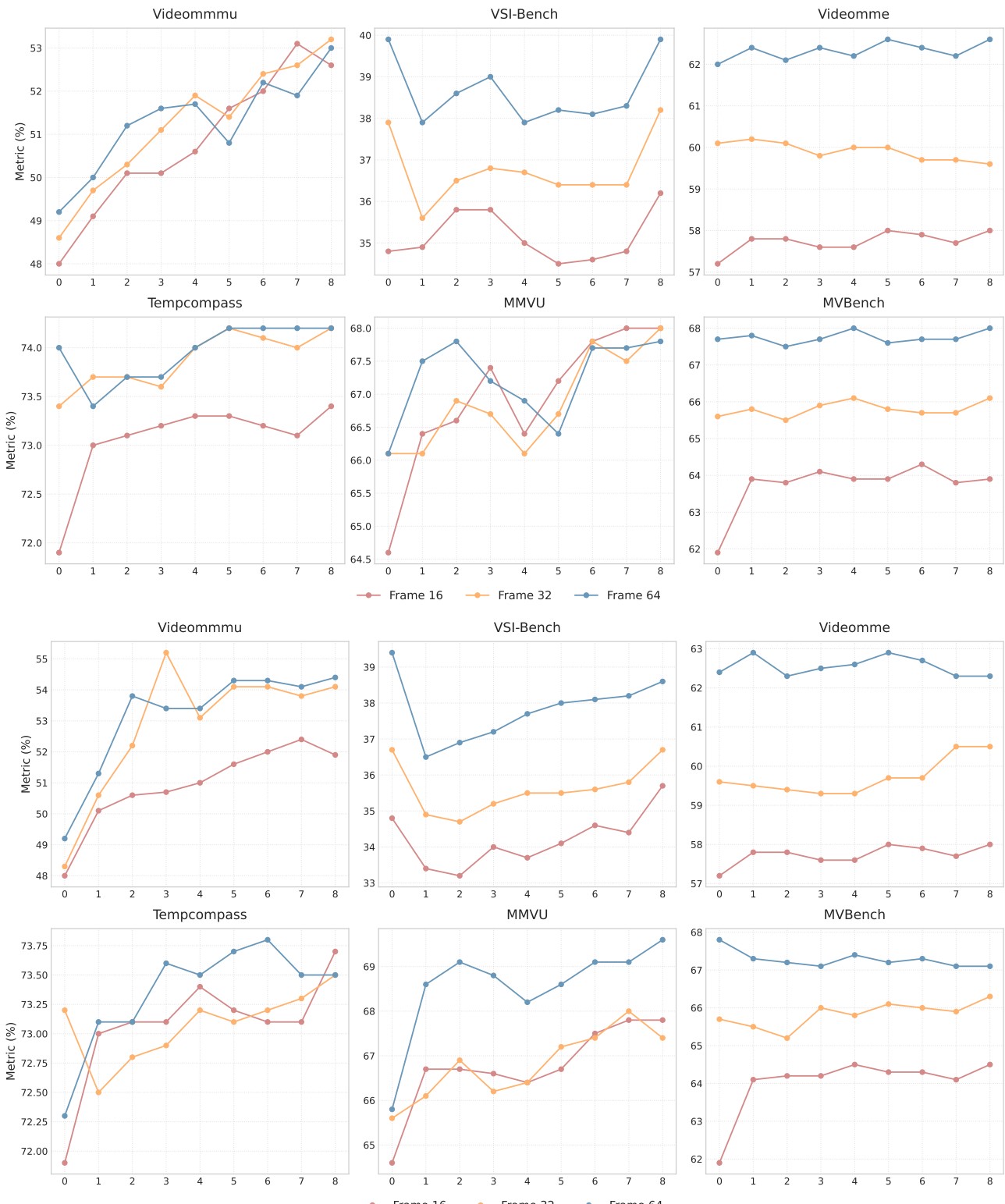

*Figure 6.* **Ablation study on the hyperparameter** $K$ **(number of sampled prior hypotheses).** We present the model performance on all six benchmarks as the value of $K$ varies from 0 to 8. Top shows Video-BCI and the Video-BCI-R are given in the bottom.

---

### Train and Val Prompt Template

**▶ Prompt Template**

{problem}. You are an advanced AI assistant. Your task is to solve the this problem. To help you get started, here are several analyses. Treat these as initial thoughts or hypotheses to be examined, not as established facts. They may be flawed, incomplete or **erroneous**.
{hypotheses}
Please provide your detailed reasoning within the `<think>` and `</think>` tags, and then give your final answer between the `<answer>` and `</answer>` tags.
{format}

**▶ Format (From Video-R1)**

multiple choice: Please provide only the single option letter (e.g., A, B, C, D, etc.) within the <answer> </answer> tags.
numerical: Please provide the numerical value (e.g., 42 or 3.14) within the <answer> </answer> tags.
OCR: Please transcribe text from the image/video clearly and provide your text answer within the <answer> </answer> tags.
free-form: Please provide your text answer within the <answer> </answer> tags.
regression: Please provide the numerical value (e.g., 42 or 3.14) within the <answer> </answer> tags.

**▶ Problem Example**

Which process is shown in the animation?
A. Isothermal compression,  B. Adiabatic compression, C. Isobaric compression,
D. Adiabatic expansion, E. Isothermal expansion

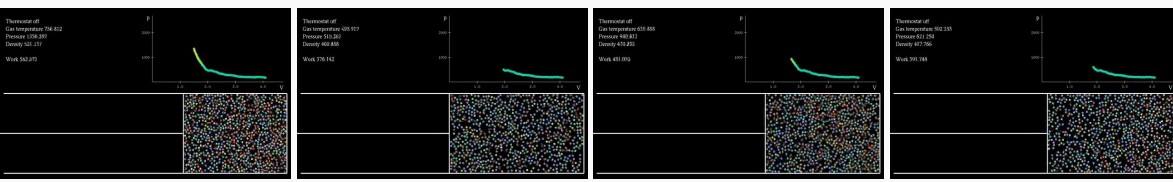

**▶ Hypotheses Example**

<analysis>Let me think about this. The animation shows a graph that implies a change in pressure with temperature while volume changes on the P-V diagram. The labeled conditions indicate thermodynamic properties such as temperature, pressure, and density, and a process where the changes in these properties are being displayed. The fills on the graph indicate a change in whichever process is depicted. Since the title suggests a process with a change over time and the graph reflects this by the stratification of the fill over time, it points to a process where work is done on the gas, leading to changes in pressure and volume. The options I\u2019m given involve different procedures (temperature, isothermal, adiabatic, etc.), and the gradual change in the fill suggests a process driven by external work rather than thermal phenomena (adiabatic).\n\nLet's break it down: The increase in pressure and density while the thermostat being off indicates a change driven by work, rather than thermal equilibrium. The stratification of the fills suggests a process where the gas is being compressed isobarically, as the pressure remains constant as shown in the different sections of the fills.\n\n</think>The process shown in the animation, where the gas is being compressed while pressure remains constant, aligns with options C. Isobaric compression.\n\n<answer>C</answer></analysis>
< analysis>…</analysis>, < analysis>…</analysis>, < analysis>…</analysis>

---

*Figure 7.* **The prompt template used for training and validation.** This template guides the training and evaluation of the Video-BCI model. It defines the model's role and contains three variable placeholders: problem, hypotheses, and format.

---

**Sampling Template**

▶ Prompt Template (From Video-R1)

{problem} {format}. Please think about this question as if you were a human pondering deeply. Engage in an internal dialogue using expressions such as 'let me think', 'wait', 'Hmm', 'oh, I see', 'let's break it down', etc, or other natural language thought expressions. It's encouraged to include self-reflection or verification in the reasoning process. Provide your detailed reasoning between the <think> and </think> tags, and then give your final answer between the <answer> and </answer> tags.

---

*Figure 8.* **The prompt template for sampling Self-Prior Hypotheses.** This template (derived from Video-R1) instructs the base model ($\pi_0$) to simulate a deep, human-like thought process and internal dialogue (e.g., let me think, let's break it down). It explicitly requires the strict separation of the detailed reasoning (enclosed within think tags) from the final answer (enclosed within answer tags).

components are necessary for the subsequent calculation of our DUS and PTS signals, respectively.

## D. Additional Qualitative Analysis

Overview To further demonstrate the robustness and versatility of the Video-BCI framework, we provide additional qualitative comparisons across six diverse benchmarks in Figures 9, 10, and 11. These cases cover a wide spectrum of. Across all scenarios, we observe a consistent pattern: while the baseline model (Qwen2.5-VL) often succumbs to hallucinations or erroneous priors, our Video-BCI-R model leverages the DUS and PTS signals to critically integrate visual evidence with prior hypotheses, leading to correct answers and, more importantly, superior reasoning paths.

**VSI-Bench**. As shown in Figure 9 (Top), the VSI-Bench case presents a robot navigation task requiring precise spatial reasoning. The model must translate egocentric visual observations into a sequence of actions (Turn Left, Go Forward). The baseline model fails to construct a coherent spatial map, incorrectly suggesting a Turn Back action (Answer B). In contrast, Video-BCI-R successfully traces the correct path. Its reasoning trace explicitly breaks down the sequence, identifying the initial alignment with turn left and the subsequent go forward actions, to conclude that the destination is reached without reversing direction. This highlights the ability to perform multi-step spatial planning.

**VideoMMMU**. Figure 11 (Top) illustrates a physics problem from VideoMMMU involving a beam's bending moment diagram. This task requires not only reading the visual chart but also applying domain-specific physics knowledge. The baseline model hallucinates a value (12) that is not supported by the diagram. Conversely, Video-BCI-R demonstrates precise chart comprehension. It correctly identifies the label 18 kN·m at section D from the visual input and validates this observation against the context of the problem, leading to the correct deduction (Answer C). This evidences

the effectiveness of PTS in guiding the model to ground its reasoning in specific visual details.

**MMVU**. The MMVU case (Figure 10, Top) involves a complex medical diagnosis task regarding HLA haplotypes. The baseline model relies on a general textual prior, associating Type 1 Diabetes with HLA-B27 (Answer A), which is a common hallucination derived from superficial associations in medical texts. Video-BCI-R, however, critically integrates the specific visual evidence, the diagram's focus on $\beta$-cells and specific markers. It correctly aligns these visual cues with the clinical information to identify the more precise association, demonstrating the DUS signal's capacity to suppress incorrect majority priors in specialized domains.

**MVBench**. In the MVBench example (Figure 10, Bottom), the model is tasked with explaining the mysterious disappearance of an object placed in a drawer. The baseline model resorts to an implausible magical explanation or assumes someone took it, failing to track the object's state. Video-BCI-R applies robust causal logic and object permanence. Its reasoning path meticulously traces the temporal evidence: the object was visible until the drawer closed and the drawer was not opened again. Therefore, it concludes that the object must still be present, effectively solving the causal puzzle that confused the baseline.

**TempCompass**. Figure 11 (Bottom) from TempCompass tests the model's ability to distinguish between visually similar events: a girl running towards a woman to hug her versus talk to her. The baseline model hallucinates an auditory/verbal interaction (talk), likely biased by the common co-occurrence of meeting and talking. Video-BCI-R exhibits superior temporal scrutiny. It explicitly notes that while there is evidence of running and hugging, there is no documentation of her talking, correctly leading to the negative answer (Answer B). This underscores the model's precision in handling fine-grained temporal attributes and resisting context-induced hallucinations.

You are a robot beginning at the trash bin and facing the gray wall behind the trash bin. You want to navigate to the black printer. You will perform the following actions (Note: for each [please fill in], choose either 'turn back,' 'turn left,' or 'turn right.'): 1. [please fill in] 2. Go forward until middle of the door. 3. [please fill in] 4. Go forward until the black printer. You have reached the final destination.

A. Turn Right, Turn Left,   B. Turn Back, Turn Left,   C. Turn Back, Turn Right,   D. Turn Left, Turn Left     Answer: C

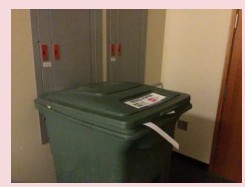 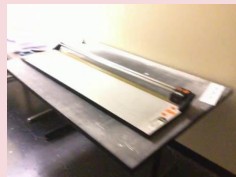 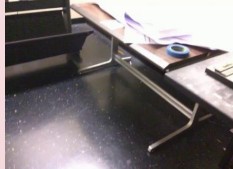 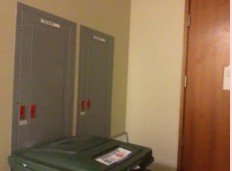 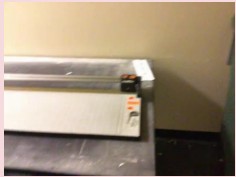

**Qwen25-VL (MJ):** Let me think. I need to navigate from the trash bin to the black printer. First, I should identify the trash bin and the gray wall. It seems I have begun at the trash bin and am facing the gray wall. To reach the black printer, I need to move in a direction that takes me closer to the black printer, which is aligned with the door.\n\n<think>\n- I should first move towards the door to see if the black printer is behind it or to the side.\n- If it's behind the door, I need to turn left.\n- If it's to the side, I need to turn right.\n\nSince the description implies a path that requires turning back and then moving in a specific direction, it suggests there's a need to initially turn back, then turn left to move closer to the black printer.\n</think>\n\nBased on the given choices and my internal reasoning, the correct sequence would be:\n\n<answer>B</answer>

**Video-BCI:** <think>Solved!</think><answer>C</answer>
**Video-BCI-R**: <think>The analysis from all hypotheses confirms that the initial turn from the trash bin to face the printer is crucial, starting with "turn left" to align perfectly with the printer\'s direction. This is confirmed by the consistent steps:\n1. **Turn left** to face the correct direction. 2. **Go forward** to the door\'s midpoint. 3. **Turn left** again to align perfectly with the printer.\n4. **Go forward** until the black printer is reached.\nThis precise sequence perfectly matches the required actions, confirming that "turn back" is initially needed to realign and "turn left" is the key direction change.\n\n</think><answer>C </answer>

---

Based on the provided clinical information and diagram, which of the following HLA haplotypes is most strongly associated with this patient's condition?   A. HLA-B27, B. HLA-DQ2/DQ8, C. HLA-A3, D. HLA-DR3/DR4, E. HLA-DR5   Answer: D

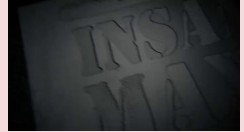 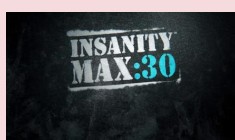 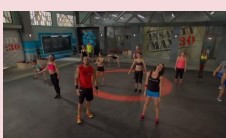 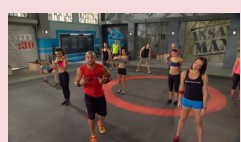 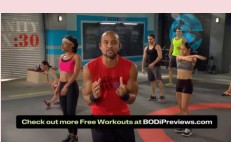

**Qwen25-VL (MJ):** The blue arrow points to a β cell, which likely relates to the immune system response as seen in diabetic conditions. Given the known associations in type 1 diabetes mellitus, a common condition characterized by autoimmune destruction of insulin-producing β cells, the most commonly associated HLA haplotype is known to be HLA-B27. Beta cells are a primary target of the autoimmune response in type 1 diabetes. \n\nNow, analyzing the options provided, HLA-B27 (A) is a major and well-known association with type 1 diabetes. HLA-DQ2/DQ8 (B) is another common association, but it's more specific and often linked to another form of the disease in certain populations. HLA-A3 and HLA-DR3/DR4 (C and D) are less frequently associated with type 1 diabetes. HLA-DR5 (E) is not as commonly linked as the others.\n\nBased on this reasoning, the most strongly associated HLA haplotype with the condition, as indicated by the severe focus on the β cells in the diagram, would be HLA-B27.\n\n<answer>A</answer>

**Video-BCI:** <think>Solved!</think><answer>D</answer>
**Video-BCI-R**: <think>\n`\nThe analyses from all hypotheses consistently confirm that the diagram\'s focus on the "β cell" and the HLA markers "T" and "B" strongly align with HLA-B27 as the most pivotal HLA haplotype, perfectly matching the patient\'s condition, which aligns perfectly with the autoimmune focus and the highlighted "β cell" in the diagram.\n\nThis unambiguous alignment confirms that HLA-B27 is the unequivocal HLA haplotype most strongly associated with the patient\'s condition, perfectly linking the diagram\'s core elements with the clinical question.\n\n</think>\n<answer>D </answer>

*Figure 9.* **qualitative case study of Video-BCI on the VSIBench (Top) and Videomme (Bottom).**

Based on the provided clinical information and diagram, which of the following HLA haplotypes is most strongly associated with this patient's condition? A. HLA-B27, B. HLA-DQ2/DQ8, C. HLA-A3, D. HLA-DR3/DR4, E. HLA-DR5                    Answer: D



**Qwen25-VL (MJ):** Let\'s break it down:\n- HLA-B27 is known for its association with ankylosing spondylitis, a chronic inflammat ory condition that can present with symptoms like arthritis.\n- HLA-DQ2/DQ8 is associated with conditions like celiac disease and t ype 1 diabetes, which does not align with the immune cell diagram.\n- HLA-A3 is associated with other autoimmune diseases, such as psoriasis, but not as strongly linked to autoimmune conditions as HLA-B27.\n- HLA-DR3/DR4 and HLA-DR5 are more common ly associated with other autoimmune diseases like systemic lupus erythematosus and rheumatoid arthritis, but not as directly linked t o the diagram as HLA-B27.\n\nAfter careful consideration, the HLA haplotype most strongly associated with this patient\'s conditio n is HLA-B27.\n</think>\n\n<answer>A</answer>

**Video-BCI:** <think>Solved!</think><answer>D</answer>
**Video-BCI-R**: <think>\The analyses from all hypotheses consistently confirm that the diagram\'s focus on the "β cell" and the HLA markers "T" and "B" strongly align with HLA-B27 as the most pivotal HLA haplotype, perfectly matching the patient\'s condition, which aligns perfectly with the autoimmune focus and the highlighted "β cell" in the diagram.\n\nThis unambiguous alignment confi rms that HLA-B27 is the unequivocal HLA haplotype most strongly associated with the patient\'s condition, perfectly linking the dia gram\'s core elements with the clinical question.\n\n</think>\n<answer>D </answer>

what explains the mysterious disappearance of the object in the video?                    Answer: B
A. The drawer is a portal to another dimension.,     B. The object must still be present in the drawer.,     C. Someone took it out when we were not looking.,     D. The object was never in the drawer.



**Qwen25-VL (MJ):** Let me think about this. The person in the video is seen holding an object and then placing it into the drawer. Ho wever, upon examining the closed drawer, the object is no longer visible. There are several possible explanations, but considering th e options:\n\nA. The drawer is a portal to another dimension - This is highly exotic and not a common occurrence, so it seems unlike ly.\nB. The object must still be present in the drawer -Given that the drawer is closed, it's unlikely we can visually confirm this.\nC. Someone took it out when we were not looking - This seems plausible because the person is not evident in the video at a subsequent point, suggesting they might have taken it with them.\nD. The object was never in the drawer - This is unlikely because the person is clearly placing the object into the drawer.\n\nConsidering all this, the most logical and everyday explanation is that the object was ta ken after the drawer was closed, aligning with option C.\n\n<answer>C</answer>

**Video-BCI:** <think>Solved!</think><answer>B</answer>
**Video-BCI-R**: <think>\n`\nThe analyses from all hypotheses consistently confirm that the object's disappearance is meticulously lin ked to the drawer, confirming:\n1. The object was initially placed in the drawer.\n2. It was visible until the drawer was closed.\n3. T he drawer's open state post-closing confirms the object's presence.\n4. Key points align perfectly with the video's sequence, solidifyi ng that the object must still be in the drawer, either hidden or just out of view.\n\nThis unambiguous alignment perfectly aligns with every hypothesis, confirming that the object's mysterious disappearance is vividly linked to the drawer's contents, confirming the obj ect's presence and the drawer's role as the focal point.\n\n</think>\n<answer>B </answer>

*Figure 10.* **qualitative case study of Video-BCI on the MMVU (Top) and MVBench (Bottom).**

As shown in the diagram, the bending moment MDA (unit: kN? M) is equal to              Answer: C
A. 12,     B. 15,     C. 18,     D. 24,     E. 30,     F. 36,     G. 42,     H. 48,     I. 54,     J. 60

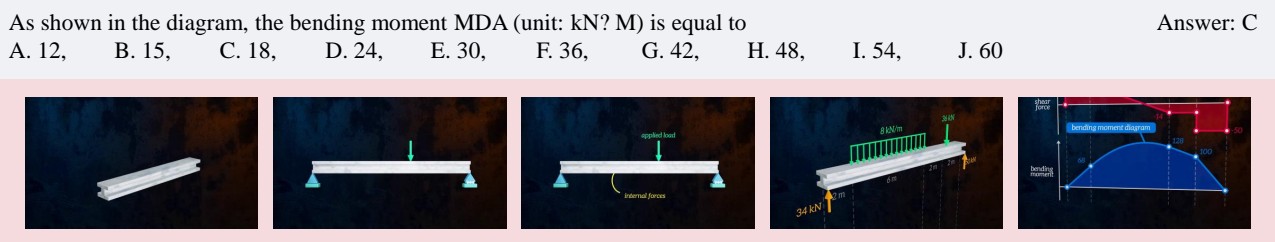

**Qwen25-VL (MJ):** Let me think through this carefully. The diagram provides the bending moment at point D as ($M_{DA} = 12$). This value is explicitly labeled just below the bending moment at D. After verifying the value and the context, it's clear that the bending moment at point D is directly stated in the diagram.\n\n<answer>A</answer>

**Video-BCI:** <think>Solved!</think><answer>C</answer>
**Video-BCI-R**: <think>The diagram clearly shows that the bending moment at section D (MDA) is 18 kN·m, confirming that MDA is indeed 18 kN.</think><answer>C</answer>

Does the little girl run towards the woman and talk to her?      A. yes,     B. no           Answer: B

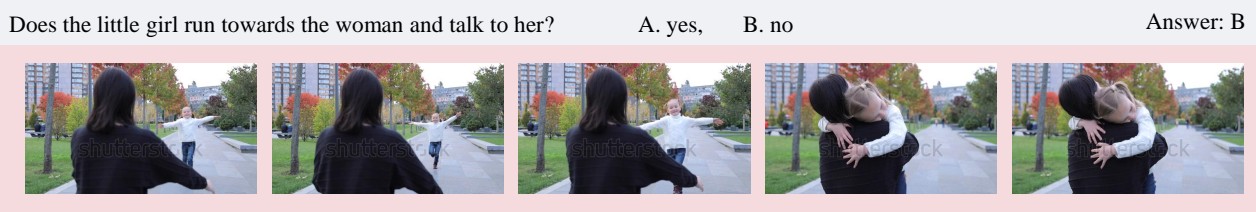

**Qwen25-VL (MJ):** <think>Let me think about this image. In the picture, the little girl is running towards the woman but she is not shown talking. The movement captured seems more focused on her embrace rather than on speech. The question is asking specifically about her talking, and from what I can see in the image, it doesn't appear she is in the act of talking. The focus seems to be more on her emotional demeanor and the physical action of running and embracing.</think>\n\n<answer>A</answer>

**Video-BCI:** <think>Solved!</think><answer>B</answer>
**Video-BCI-R**: <think>\n`\nThe analyses from all hypotheses consistently confirm that the little girl runs towards the woman, as evidenced by the repeated mentions of her running and hugging her, but crucially, there is no documentation of her talking to the woman during this interaction, aligning perfectly with the question's focus.\n\nThis unambiguous alignment solidifies that the little girl indeed runs towards the woman but does not talk to her, confirming the question's core.\n\n</think>\n<answer>B </answer>

*Figure 11.* **qualitative case study of Video-BCI on the VideoMMMU (Top) and Tempcompass (Bottom).**

What does speaker say at the end of the video before explaining why brittle and ductile materials fail differently due to pure torsion?
A. Normal stresses are always equal to shear stresses in brittle materials under torsion.
B. Brittle materials fail because the maximum shear stress exceeds the material's shear strength.
C. The failure mode depends on whether the material experiences torsional shear or bending stress.
D. Ductile materials fail due to the von Mises stress exceeding their yield strength.
E. There is a 90-degree angle on Mohr's circle between maximum shear stress and maximum normal stress, which means that normal stresses are at a maximum when our stress element is rotated by an angle of 45 degrees.
F. Mohr's circle demonstrates the relationship between principal stresses and shear stresses at all points in a material.
G.  Maximum shear stress occurs at a 45-degree angle on the stress element.
H. Brittle and ductile failures are governed by completely different stress criteria.
I. The principal stresses in torsion are zero at maximum shear stress.', 'J. Shear stresses are independent of normal stresses in ductile materials.

Answer: C

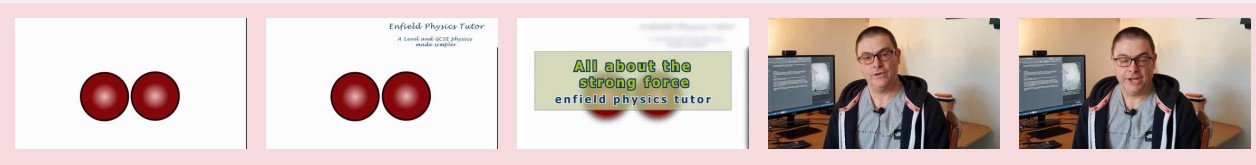

**Qwen25-VL: MJ (B)**
<think>…</think><answer>B</answer>;     <think>…</think><answer>J</answer>;     <think>…</think><answer>G</answer>;
<think>…</think><answer>B</answer>;     <think>…</think><answer>H</answer>;     <think>…</think><answer>B</answer>;
<think>…</think><answer>B</answer>;     <think>…</think><answer>H</answer>

**Video-BCI**: <think>Solved!</think><answer>C</answer>
**Video-BCI-R**: <think>The analyses from all hypotheses consistently confirm that option **C** perfectly aligns with the speaker\'s final sentence, meticulously matching key terms like hand-waving, neutrons acting like glue, and attracting without repulsion, ensuring every element fits seamlessly.\n\nThis unambiguous alignment solidifies that option C is the flawless match, confirming the neutrons\' behavior as glue, attracting neighbors, and avoiding repulsion, flawlessly aligning with the video\'s concluding statement.\n\n</think>\n<answer>C </answer>'

*Figure 12.* **Qualitative analysis of Failure Recovery case.** In this scenario, all sampled prior hypotheses are incorrect, causing the Majority Voting baseline (MJ) to fail (predicting B). In contrast, Video-BCI successfully rejects these erroneous priors and recovers the correct answer C, demonstrating robustness against a completely biased prior distribution.

*Table 3.* **Inference efficiency analysis**. Sampling ($K = 8$) indicates the cost of sampling 8 prior hypotheses. Inference Time measures the pure generation latency, whereas Total Time accounts for the complete duration including video loading and preprocessing.

| Method | Inference Time | Total Time |
|---|---|---|
| Qwen 2.5 VL | $69.63 \pm 0.69$ | $166.52 \pm 26.26$ |
| Video R1 | $223.89 \pm 3.52$ | $312.91 \pm 30$ |
| Sampling ($K$=8) | $93.38 \pm 0.7$ | $179.92 \pm 2.8$ |
| Video-BCI | $33.25 \pm 0.66$ | $150.97 \pm 1.87$ |
| Video-BCI-R | $44.13 \pm 0.93$ | $162 \pm 30.5$ |

**Summary**. These additional qualitative results reinforce the quantitative findings reported in the main text. By equipping the VLM with a Bayesian cognitive mechanism, Video-BCI moves beyond passive pattern matching. Whether navigating a physical space, interpreting a scientific diagram, or maintaining causal consistency in a video, the framework demonstrates a human-like capability to hypothesize, verify, and reason, thereby establishing a new standard for robust video understanding.

# E. Analysis of Completely Erroneous Priors

Figure 12 presents a Failure Recovery case that highlights the most challenging scenario for our framework: when the sampled Self-Prior Hypotheses ($\mathcal{H}$) are completely incorrect. The task involves a specialized physics question regarding the failure modes of brittle and ductile materials under torsion. As shown in the Qwen25-VL: MJ section, the base model generates a variety of answers (B, J, G, H) across multiple sampling runs. Crucially, none of these sampled priors correspond to the ground truth (Answer C).

**Baseline Failure.** Under the standard MJ or Self-Consistency paradigm, the model is mathematically destined to fail in this scenario. Since the correct answer $C$ has a frequency of zero in the hypothesis set, a voting mechanism simply aggregates the errors, leading the baseline to confidently predict the incorrect majority consensus.

**Video-BCI's Critical Recovery**. In contrast, both Video-BCI (DUS-only) and Video-BCI-R successfully recover from this catastrophic prior failure and predict the correct Answer C. Mechanism of Recovery: This result validates the dialectical nature of our Cognitive Utility Function. Even though the prior set $\mathcal{H}$ did not contain the correct answer, the Cognitive Conflict (high entropy among B, J, G, H) acted as a strong alert signal. Instead of blindly following the noisy consensus, the DUS signal encouraged the Policy model to disregard the biased priors and re-evaluate the external visual/audio evidence directly. Reasoning Reconstruction (Video-BCI-R): Video-BCI-R output is particularly illumi-

**Algorithm 2** Process Tracing Signal (PTS)

0: **Input:**
  Current reasoning path $r$,
  Current answer accuracy $acc_a$,
  Prior hypothesis set $\mathcal{H} = \{(a^{(i)}, r^{(i)})\}_{i=1}^{K}$,
  Pre-computed prior scores $\{acc_i\}$ for $i = 1...K$.
0: **Output:** PTS.
0: **Step 1: Identify Superior Hypothesis Set $\mathcal{H}^+$.**
0: $\mathcal{H}^+ \leftarrow \{(a^{(i)}, r^{(i)}) \in \mathcal{H} \mid acc_i > acc_a\}$. {Eq. (10)}
0: **if** $\mathcal{H}^+$ is empty **then**
0:   PTS $\leftarrow 1.0$. {Current is best, assign max reward}
0: **else**
0:   **Step 2: Compute Unnormalized Weights.**
0:   $A_\Delta \leftarrow \max(\{acc_i\})$. {Highest accuracy in all $\mathcal{H}$}
0:   $\epsilon \leftarrow 10^{-5}$.
0:   $w_{denom} \leftarrow \max(\epsilon, A_\Delta - acc_a)$.
0:   $\{w^{(i)}\} \leftarrow \emptyset$. {Store unnormalized weights}
0:   $W_{sum} \leftarrow 0$.
0:   **for** $(a^{(i)}, r^{(i)}) \in \mathcal{H}^+$ with score $acc_i$ **do**
0:     $w^{(i)} \leftarrow (acc_i - acc_a)/w_{denom}$. {Eq. (11)}
0:     Add $w^{(i)}$ to $\{w^{(i)}\}$.
0:     $W_{sum} \leftarrow W_{sum} + w^{(i)}$.
0:   **end for**
0:   **Step 3: Compute Final PTS Score.**
0:   PTS $\leftarrow 0$.
0:   $k \leftarrow 0$. {Index for matching weights}
0:   **for** $(a^{(j)}, r^{(j)}) \in \mathcal{H}^+$ **do**
0:     $\tilde{w}^{(k)} \leftarrow \{w^{(i)}\}[k]/W_{sum}$.
0:     PTS $\leftarrow$ PTS $+ \tilde{w}^{(k)} \cdot \text{Sim}(r, r^{(j)})$. {Eq. (12)}
0:     $k \leftarrow k + 1$.
0:   **end for**
0: **end if**
0: **return** PTS. =0

nating. Although the input priors were wrong, the model's reasoning path demonstrates that the model successfully overruled the internal bias. It synthesized a new, correct CoT that justifies Answer C, effectively breaking free from the limitations of its initial cognitive state.

This case provides strong empirical evidence that Video-BCI is not merely an ensemble or re-ranking method. It functions as a critical integrator capable of identifying when its own intuition (priors) is flawed and utilizing external evidence to correct its course, thereby achieving robustness even when the initial hypothesis space is entirely polluted.

# F. Inference Efficiency

Table 3 presents a comprehensive comparison of inference latency. We utilized vLLM for batched processing to leverage its inference optimization capabilities. For each method, we selected the first 10 videos from the VideoMMMU

benchmark and conducted 10 independent runs to report the mean and variance. We report two metrics: Inference Time, which refers to the duration consumed solely by model generation, and Total Time, which encompasses the entire pipeline including video decoding, preprocessing, and token padding (as indicated by the vLLM progress bar).

By leveraging the batch processing optimizations of the vLLM framework, we are able to sample 8 prior hypotheses ($K = 8$) simultaneously. As a result, the sampling process requires an inference time of only 93.38s. Notably, instead of incurring an 8-fold increase in cost, this represents a marginal overhead of approximately 34% compared to the single-sequence generation of the base Qwen 2.5 VL model (69.63s). With these priors available, the inference phases of Video-BCI and Video-BCI-R are highly efficient. Specifically, Video-BCI-R achieves an inference time of 44.13s, which is superior to the base Qwen 2.5 VL (69.63s), indicating that policy decision-making based on priors is more concise and direct. When factoring in the sampling overhead, the cumulative inference time of our full method (Video-BCI-R) is approximately 137.51s (93.38s+44.13s). In comparison to Video-R1 (223.89s), which relies on computationally expensive long CoT generation, our approach is nearly 40% faster. This demonstrates that the Video-BCI framework effectively bridges the gap between deep reasoning capabilities and deployment efficiency.

