# OpenReview forum: "Video-BCI: Bayesian Cognitive Integration of Self-Prior Hypotheses for Video Understanding"
_ICML.cc/2026/Conference — ICML 2026 regular_

### Official Review · Reviewer_gHzk · 2026-03-12

**Soundness:** 3
**Presentation:** 3
**Significance:** 3
**Originality:** 3
**Overall Recommendation:** 5
**Confidence:** 3

**Summary:**

The paper proposes to interpret video reasoning in vision–language models as a Bayesian-style cognitive process. The authors aim to address the issue that existing video understanding models tend to behave as naive empiricists that directly map video inputs to answers without explicitly modeling competing hypotheses. To address this, the paper introduces two signals (DUS and PTS) to encourage multi-hypothesis reasoning and improve robustness in video understanding tasks. Experiments are conducted on several video benchmarks (e.g., MVBench, VideoMMMU, MMVU), showing consistent improvements over recent RL-based baselines such as Video-R1.

**Compliance With Llm Reviewing Policy:**

Affirmed.

**Final Justification:**

I will keep my score at the current level or strive to improve it.

**Key Questions For Authors:**

1. The Bayesian formulation appears largely conceptual. Could the authors clarify whether the framework performs any explicit Bayesian inference or posterior update beyond heuristic sampling?
2. What is the additional computational cost compared to existing RL-based methods such as Video-R1?

**Limitations:**

The method is conceptually motivated and empirically validated, but the Bayesian interpretation appears more heuristic than a rigorous probabilistic formulation, which somewhat limits the theoretical contribution.

**Strengths And Weaknesses:**

Strengths
1. The paper introduces an interesting perspective by interpreting video reasoning as a Bayesian cognitive process. This framing provides an intuitive explanation for why multi-hypothesis reasoning may help improve robustness in video understanding tasks.
2. The two proposed signals (DUS and PTS) are conceptually well motivated. DUS encourages models to challenge incorrect majority hypotheses, while PTS performs a form of reasoning-path distillation from stronger hypotheses.
3. The authors evaluate the method on six benchmarks covering both reasoning-focused and general video understanding tasks. The results show consistent improvements compared to several recent RL-based baselines.

Weaknesses
1. Although the method is framed as a Bayesian cognitive process, the implementation does not appear to perform explicit Bayesian inference or posterior updating. Instead, the “priors” are sampled from a base model and integrated via reinforcement learning. As a result, the Bayesian interpretation appears mostly conceptual rather than mathematically grounded.
2. While the results are consistent across benchmarks, the improvements over strong RL baselines are relatively modest (typically around 3–4%). It remains unclear whether these gains justify the additional conceptual complexity introduced by the framework.

---

> ### Author Rebuttal · Authors · 2026-03-31
>
> We sincerely thank the reviewer for the constructive feedback. We are highly encouraged that you find our perspective of reframing video reasoning as a Bayesian cognitive process interesting, and that the intrinsic signals (DUS and PTS) are conceptually well-motivated. Regarding your specific questions, we provide detailed responses below:
>
> ### 1. Explicit Bayesian Inference vs. Heuristic Sampling (Q1 & W1)
> Regarding the current implementation, we completely agree that we adopted reinforcement learning (RL) techniques rather than performing a strict, closed-form Bayesian posterior update. We will make this distinction clearer in the revised manuscript to `avoid overstating theoretical contributions`.
>
> However, we wish to clarify that framing this work within a Bayesian context is not merely a heuristic wrapper; it serves as the `strict mathematical foundation` for deriving our objective functions. For massive Vision-Language Models (VLMs), calculating a strict Bayesian posterior distribution (which involves integrating over the entire reasoning space) is mathematically intractable. Therefore, our framework constructs an `empirical approximation` of the prior distribution through multi-hypothesis sampling. Under this approximation, DUS and PTS are not arbitrarily designed heuristic rules, but strictly correspond to:
> - Evaluating the `cognitive conflict` (i.e., information entropy) of the prior distribution.
> - Maximizing the alignment between the posterior reasoning path and the external evidence.
>
> In this context, RL acts as a `tractable optimizer` to realize the Bayesian integration. We will add a dedicated Discussion paragraph in Section 3 to transparently address this trade-off between strict Bayesian probabilistic modeling and LLM-based RL approximations.
> ### 2. Additional Computational Cost and Justification of Gains (Q2 & W2)
> Compared to existing RL-based methods (such as Video-R1), the additional computational cost introduced by Video-BCI is `surprisingly minimal`.
> - **Training Cost**: During the actual RL training iterations and optimization steps, Video-BCI `introduces no additional overhead`. Because DUS and PTS act as intrinsic reward scalars, the CPU/GPU time required for their mathematical computation is virtually negligible. The only extra computational cost occurs `prior to RL training`, specifically during the multi-hypothesis sampling phase (offline data generation). Since we only need to sample the K=8 priors once per training prompt, this process can be completed entirely offline and in parallel, taking only a small fraction of the total training time.
> - **Justification of Gains**: While an absolute improvement of 3-4% might seem modest in traditional perception tasks, it represents a `massive leap forward` on extremely challenging, reasoning-heavy benchmarks like MMVU and VideoMMMU (for instance, some researchers heavily rely on large-scale instance annotations merely to achieve a ~1% performance boost). These benchmarks require deep causal logic, an area where previous models have plateaued. Achieving SOTA performance without the need for expensive manual CoT annotation data, and doing so with almost the `same training wall-clock time` as standard GRPO, makes the conceptual complexity and the minimal sampling overhead highly worthwhile.
>
> We thank you again for your time, support, and constructive feedback, which greatly helps us refine the theoretical boundaries of this work. If you have any further questions, please feel free to contact us.
>
> Sincerely,
>
> The Authors

---

> > ### Author Rebuttal · Reviewer_gHzk · 2026-04-02
> >
> > I will keep my score at the current level or strive to improve it.

---

> > > ### Author Response · Authors · 2026-04-06
> > >
> > > We sincerely thank the reviewer for participating in the peer-review process and for conducting a thorough and rigorous evaluation of our manuscript. We greatly appreciate the scholarly expertise and meticulous dedication demonstrated throughout your assessment. Accordingly, we will carefully address each of your comments and incorporate the suggested revisions, ensuring that the final version presents more robust and reproducible findings.
> > >
> > > Once again, we are deeply grateful for your valuable contribution to the scholarly community and for your constructive engagement with our work.
> > >
> > > Sincerely,
> > >
> > > The Authors

---

### Official Review · Reviewer_4KWX · 2026-03-13

**Soundness:** 3
**Presentation:** 3
**Significance:** 3
**Originality:** 2
**Overall Recommendation:** 4
**Confidence:** 4

**Summary:**

This paper proposes Video-BCI, which models the video understanding task as a Bayesian cognitive process. It utilizes a base model as prior knowledge (sampling a set of prior hypotheses from itself), and the training process optimizes the inference path through DUS (dialectical uncertainty signal) and PTS (process tracking signal). The main contribution of this framework is that it draws on the hypothesis testing and reflective reasoning of human cognition to achieve a combination of passive priors and active reasoning in large models.

**Compliance With Llm Reviewing Policy:**

Affirmed.

**Final Justification:**

The rebuttal successfully addressed several of my main concerns, particularly regarding methodological clarity, efficiency trade-offs, and the role of different components. It improved my understanding of the framework and increased my confidence in both the technical soundness and practical value of the approach. While some concerns (e.g., broader empirical validation and clearer positioning of novelty) are only partially resolved, they do not fundamentally undermine the contribution.

As a result, I am increasing my score.
Overall, I find the work to be solid, with clear potential impact, and I encourage the authors to incorporate the rebuttal clarifications into the final version to further strengthen the paper.

**Key Questions For Authors:**

See Strength and Weakness.Experimentation is very important. If authors can provide stronger evidence and more comprehensive experiments, I would be appreciate to raise the score.

**Limitations:**

Yes

**Strengths And Weaknesses:**

Strength:
1. This paper combines Bayesian cognitive processes with reinforcement learning signals (DUS, PTS), which is a relatively novel idea.
2. The complementary signal design of DUS and PTS is quite innovative. DUS encourages the model to move beyond prior knowledge, while PTS distills high-quality inference processes.
3. The framework can adapt to different downstream tasks. For example, the computation of DUS can be performed in different ways depending on whether the problem is classified or not.

Weakness:

1. Intuitively, the framework is heavily influenced by the base model (which dominates the generation of prior knowledge, crucial for the overall reasoning and optimization process). However, this paper only uses Qwen2.5 as the base model. Did the authors conduct experiments with other base models? Or can they prove that the base model has little impact on the framework?
2. The framework needs to generate K prior hypotheses for each sample. Although the appendix points out that it is more efficient than methods with very long thought chains, the efficiency of this framework remains a concern as K increases.
3. From the ablation experiments, PTS and DUS seem to have little impact on the model (in Table 2, there is little difference in model performance between adding only DUS/PTS and adding both; in fact, using only DUS might even outperform adding both in the video reasoning benchmark). Can the necessity of these two signals be proven in other ways?
4. An ablation experiment was conducted on the hyperparameter K in the prisoner model, but the results show that the selection of K does not seem to have much impact on the model performance (Figure 6). In addition, the authors recommend a value of 4 for K. Is this really sufficient for the calculation of DUS? (The main purpose of DUS is to encourage the model to generate results that are different from the majority of prior inferences. Intuitively, it is thought that K should be larger.)

---

> ### Author Rebuttal · Authors · 2026-03-31
>
> We sincerely thank the reviewer for the positive evaluation of our paper. Regarding your questions on experimental completeness and methodological details, we provide detailed responses below:
>
> ### 1. Base Model Dependency and Impact (W1)
> The base model indeed determines the lower bound of prior knowledge. In our experimental setup, we currently only evaluate Qwen2.5-VL to ```maintain base architecture consistency``` with the most cutting-edge RL video models (such as Video-R1 and GRPO-CARE), thereby ensuring a fair comparison. Given the massive computational overhead of RL training for long videos, we concentrated our resources on ```this established SOTA baseline``` during the submission period. We commit to extending Video-BCI to other mainstream model families, such as LLaVA, in the final version. From a theoretical perspective, Video-BCI is universally applicable to different base models. Its core objective is precisely to ```identify and correct biases``` inherent in the base model. The weaker the base model, the more prone it is to generate erroneous intuitive consensus, which allows our DUS (challenging erroneous consensus) signal to ```provide larger reward gradients```. This forces the policy model to learn anti-bias reasoning capabilities.
>
> ### 2. Efficiency Concerns Caused by Sampling K Hypotheses (W2)
> We wish to make an important clarification here. During the inference phase, although we still need to sample priors, the overall inference efficiency of Video-BCI retains a ```significant competitive advantage```. Unlike methods that force the model to output extremely long thought chains (e.g., Video-R1), the prior hypotheses generated by our base model are ```typically very short``` More importantly, the process of generating priors can be ```fully parallelized```. As shown in our inference time comparison: even including the sampling overhead, the end-to-end inference time of Video-BCI remains superior to long-reasoning baselines. This is because the policy model can draw conclusions directly and swiftly after integrating these information priors, eliminating the need for a lengthy reasoning process. Although increasing K introduces some sampling overhead, under our recommended settings, Video-BCI achieves an ```excellent speed-accuracy balance```.
>
> ### 3. Necessity of Using DUS and PTS Jointly (W3)
> In reinforcement learning, the combination of multiple reward signals often encounters a certain trade-off, which explains why the numerical accuracy gain of their combination does not appear to be strictly additive. The necessity of both signals lies in their ```complementary cognitive functions```:
>
> - **DUS is responsible for breaking biases:** It determines whether the model can choose the correct, counter-intuitive path among multiple options. Therefore, on general video benchmarks leaning towards perception and common sense, ```using DUS alone yields``` substantial benefits.
>
> - **PTS is responsible for logical rigor:** It determines whether the model can provide tightly integrated temporal and causal reasoning chains after selecting the correct answer.
>
> Although numerical differences are not significant on some benchmarks, the Qualitative Analysis provided in our Appendix clearly shows that: when DUS and PTS are used jointly, the generated Reasoning Trajectory possesses ```extremely high causal coherence``` and quality. The reliability of this reasoning process is indispensable for deploying large models in complex tasks.
>
> ### 4. Clarification on Inference Mechanism and Hyperparameter K (W4 & W2)
> We wish to clearly distinguish the K value settings between the training and inference phases: as stated in our ablation study, we recommend ```using K=4 during inference```, but during the RL training phase, we uniformly sample ```K=8 to compute signals``` for DUS/PTS. Regarding your intuition that K should be larger when calculating DUS, this is absolutely correct in a strict statistical sense. However, in the specific practice of video understanding, K=8 is completely sufficient and reasonable for the following reasons:
> - **Saturation of option space:** Video understanding benchmarks (like MMVU) are mostly 4-option multiple-choice questions. Within an option space of only 4 choices, sampling K=8 is more than enough to expose the base model's cognitive biases and generate ```significant distribution imbalances```. This conflict is sufficient for DUS to accurately calculate information entropy and ```capture the minority```.
>
> These are our responses to your questions regarding Video-BCI. We hope these explanations clarify your concerns. If you have any further questions, please do not hesitate to contact us.
>
> Sincerely,
>
> The Authors

---

> > ### Author Rebuttal · Reviewer_4KWX · 2026-04-05
> >
> > We sincerely thank the authors for their response, which has to some extent addressed our concerns regarding the foundation model. We will increase our score and hope that the authors can provide a more detailed explanation in the final version.

---

> > > ### Author Response · Authors · 2026-04-06
> > >
> > > Dear Reviewer,
> > >
> > > Thank you for your positive feedback and constructive suggestions. We are glad that our response has helped alleviate your concerns regarding the foundation model. We sincerely appreciate your decision to raise your score, which is highly encouraging to us. Regarding your suggestion to provide a more detailed explanation in the final version, we are currently conducting additional experiments and analyses to further substantiate our findings. The corresponding results, along with comprehensive discussions, will be fully integrated into the final manuscript.
> > >
> > > We once again thank you for your professional, thorough, and insightful review.
> > >
> > > Sincerely,
> > >
> > > The Authors

---

### Official Review · Reviewer_Pg9K · 2026-03-13

**Soundness:** 3
**Presentation:** 2
**Significance:** 2
**Originality:** 2
**Overall Recommendation:** 4
**Confidence:** 2

**Summary:**

This paper proposes Video-BCI, a framework that incorporates two novel metrics—Dialectical Uncertainty Signal (DUS) and Process Tracing Signal (PTS)—into the reward function. The authors appear to present an important concept by reframing video reasoning as a Bayesian cognitive process rather than a direct mapping task. Specifically, the authors intend to address the concept of enhancing a model’s visual understanding capabilities by:
1. Encouraging the model to break free from the constraints of its prior distribution to perform more extensive exploration.
2. Distilling and retaining high-quality reasoning trajectories from the prior space.

**Compliance With Llm Reviewing Policy:**

Affirmed.

**Final Justification:**

Thanks for the authors' detailed responses, which successfully addressed my concerns. I have no further questions. Since my original rating is already positive, I'd like to maintain it.

**Key Questions For Authors:**

- The base model, i.e., Qwen2.5-VL, used in this paper is a little outdated. It would further strengthen the contributions of this paper if the authors could apply the proposed method to more recent VLMs.
- Would it be possible to compare the computational overhead of Video-BCI and standard GRPO during training (e.g., wall-clock time per step, rollout count, and time spent on rollouts)?

**Limitations:**

yes

**Strengths And Weaknesses:**

Strength:
- The use of DUS and PTS as intrinsic rewards effectively replaces the need for costly manual Chain-of-Thought (CoT) annotations, creating a more scalable self-distillation pipeline.
- The authors have conducted extensive experiments, demonstrating consistent improvements across several benchmarks ranging from casual video understanding to complex causal reasoning.

Weakness:
- Generalization to Open-Ended Tasks: The training data and evaluation benchmarks used in this study (such as MMVU, MVBench, etc.) are predominantly composed of multiple-choice questions with fixed options. The manuscript does not provide sufficient evidence or discussion to show that Video-BCI can be stably trained on open-ended visual understanding tasks. Furthermore, it remains unclear whether the framework possesses adequate generalization capabilities for tasks requiring free-form responses where a "majority consensus" of priors might be harder to define or quantify.

- Interaction and Potential Conflict between DUS and PTS: In the ablation studies, both DUS and PTS individually show impressive performance gains compared to the baseline. However, when these two rewards are combined, the final performance is not always superior to using either metric in isolation (particularly on the Video Reasoning Benchmarks). This suggests a potential suppression effect or a lack of additive synergy between the two signals. I believe further discussion or empirical analysis is required to clarify the relationship between DUS and PTS and to explain why their combination does not consistently lead to better results.

---

> ### Author Rebuttal · Authors · 2026-03-31
>
> We sincerely thank the reviewer for the constructive feedback. We are deeply encouraged by your recognition of the scalability and novelty of our intrinsic reward mechanisms (DUS and PTS). We address your specific questions below:
>
> ### 1. Generalization to Open-Ended Tasks (W1)
> As described in our paper, the exact majority consensus of DUS cannot be directly applied to open-ended questions. However, we wish to clarify that our framework indeed ```possesses generalization capabilities``` for open-ended tasks. In Section 3.2 (Eq. 9), we specifically introduced a generalized formulation of DUS for non-classification/open-ended settings. When the answer space is unbounded, calculating P(a) based on exact string matching fails. Therefore, we evaluate ```relative superiority``` instead: we calculate the proportion of prior answers that the current answer surpasses (measured by metrics like ROUGE-L). If a is superior to the majority of priors, it receives a higher DUS reward. We also included such open-ended QA data in our training set. Furthermore, we provide qualitative examples of non-multiple-choice questions in the visualization section. In this open-ended example, the model integrates sampled priors and successfully outputs an accurate ```free-text answer``` like 15. This demonstrates the effectiveness of the generalized DUS in open-ended tasks.
> ### 2. Interaction and Potential Conflict between DUS and PTS (W2)
> Regarding their interaction, this is essentially a trade-off. The lack of strictly additive synergy is because their cognitive functions are both complementary and, in certain scenarios, competitive:
> - **DUS (Breadth & Exploration)**: Encourages breaking intuitive biases. This is extremely effective in general visual perception tasks, where the model merely needs a reminder to carefully observe the evidence rather than guessing blindly.
> - **PTS (Depth & Exploitation)**: Encourages rigorous logic and causal chain tracking. On benchmarks requiring only shallow reasoning, forcing the model to trace lengthy logical paths (PTS) might introduce `unnecessary noise` (over-thinking), thereby slightly suppressing the raw exploration gain from DUS.
>
> However, on reasoning-heavy benchmarks, their combination is crucial. As shown in our Qualitative Analysis (Appendix), although the numerical gains seem saturated, when DUS and PTS are used jointly, the generated Reasoning Trajectory possesses significantly higher `causal coherence and quality`. The reliability of this reasoning process is indispensable for deploying large models in complex tasks.
> ### 3. Outdated Base Model (Q1)
> We agree with your view that extending it to newer, more diverse VLMs will strengthen the paper's contribution. We chose Qwen2.5-VL primarily to `ensure strict alignment` and fair comparison with the most cutting-edge RL video baselines (e.g., Video-R1 and GRPO-CARE). Given the rapid iteration in the open-source community, we selected a widely adopted version as our foundation. Considering the massive computational overhead of video RL training, we concentrated our submission resources on penetrating this `recognized SOTA baseline`. In the final version, we will supplement extended experiments using other architectures.
> ### 4. Computational Overhead Comparison vs. Standard GRPO (Q2)
> Thank you for focusing on this operational detail. Regarding the training overhead, compared to standard GRPO, Video-BCI introduces `no additional overhead` during the actual RL training iterations. Because DUS and PTS are intrinsic reward scalars, the CPU/GPU time required for their mathematical computation is negligible. The only extra overhead occurs `prior to RL training`, during the multi-hypothesis prior sampling phase (offline data generation). Since we only need to sample the K=8 priors once per training prompt, this process can be completed entirely offline and in parallel. Therefore, during the actual RL Rollout and optimization steps, the wall-clock time per step and rollout counts are `identical to standard GRPO`.
> As an additional note regarding the inference phase overhead, please refer to Table 3 (Inference efficiency analysis). Even when including the time spent sampling priors during inference, the overall inference efficiency of Video-BCI still surpasses previous `SOTA baselines` that rely on long thought chains.
>
> We hope these clarifications address your concerns and demonstrate the robustness of our framework. Thank you again for your time and constructive feedback. If you have any further questions, please feel free to let us know.
>
> Sincerely,
>
> The Authors

---

> > ### Author Rebuttal · Reviewer_Pg9K · 2026-04-04
> >
> > Thanks for the authors' detailed responses, which successfully addressed my concerns. I have no further questions. Since my original rating is already positive, I'd like to maintain it.

---

> > > ### Author Response · Authors · 2026-04-06
> > >
> > > We sincerely thank the reviewer for participating in the peer-review process and for conducting a thorough and rigorous evaluation of our manuscript. We greatly appreciate the scholarly expertise and meticulous dedication demonstrated throughout your assessment. Accordingly, we will carefully address each of your comments and incorporate the suggested revisions, ensuring that the final version presents more robust and reproducible findings.
> > >
> > > Once again, we are deeply grateful for your valuable contribution to the scholarly community and for your constructive engagement with our work.
> > >
> > > Sincerely,
> > >
> > > The Authors

---

### Official Review · Reviewer_Ka56 · 2026-03-13

**Soundness:** 3
**Presentation:** 3
**Significance:** 2
**Originality:** 2
**Overall Recommendation:** 4
**Confidence:** 4

**Summary:**

This paper proposes Video-BCI, a post-training framework that reformulates video reasoning as a process of self-prior hypothesis sampling followed by posterior-style integration. Video-BCI first samples multiple self-prior hypotheses from a base model, and then trains a policy model to integrate these hypotheses with video evidence to generate the final answer and reasoning trace. The key contribution is a cognitive utility function with two signals: DUS for rewarding correct minority judgments, and PTS for distilling better reasoning traces from prior hypotheses. Experiments on six video benchmarks show consistent improvements over RL baselines, with especially notable gains on reasoning-heavy tasks such as MMVU and VideoMMMU.

**Compliance With Llm Reviewing Policy:**

Affirmed.

**Final Justification:**

I lean towards weak accept.

**Key Questions For Authors:**

1. I would be interested in seeing how the proposed method compares against stronger equal-budget baselines under the same K=8 hypothesis budget. In particular, it would be useful to evaluate best-of-N, self-consistency, or simple reranking using the same sampled prior hypotheses. This would clarify how much of the gain comes from the proposed RL framework versus simply having access to multiple candidate hypotheses.

2. Could the authors report multi-seed results or variance statistics for the main benchmarks? Since the method depends on prior sampling and RL training, this would help establish the stability of the reported gains.

3. Could the authors elaborate on when PTS helps and when it does not? The current results suggest that its benefit over the DUS-only variant is not always consistent across benchmarks, and a more detailed explanation would help clarify the scope of the method.

**Strengths And Weaknesses:**

## Strengths:

1. The paper is well-motivated, and the overall narrative is coherent from problem formulation to reward design.
2. The method does not rely on manually annotated reasoning traces and instead distills from the model's own prior hypotheses, which improves scalability.
3. DUS and PTS play clear and complementary roles in the overall framework.
4. The method shows strong empirical gains over RL baselines on multiple video benchmarks.

## Weaknesses:

1. The main contribution seems to be post-training reward design and multi-hypothesis integration, rather than a clear advance in video temporal modeling. Similar ideas are already related to self-consistency, reranking, and process-level distillation, so the paper should better clarify what is fundamentally new in DUS/PTS beyond these existing paradigms.
2. Although the paper is framed as a video understanding method, the evidence for improved video-specific temporal reasoning is limited. The current results do not clearly show whether the gain comes from better temporal modeling or simply better answer refinement.
3. The gains from PTS over the DUS-only variant appear somewhat inconsistent across benchmarks. The paper should better explain when PTS helps and why its benefit is not always stable.
4. The Bayesian framing feels somewhat stronger than the actual method. The approach appears closer to heuristic RL reward shaping than to a rigorous Bayesian posterior update.
5. Some key implementation details remain unclear, particularly the similarity function $Sim(r, r^{(i)})$ used in PTS. Additionally, the paper does not discuss the method’s sensitivity to the hyperparameter $K$, sampling settings, or random seeds.

---

> ### Author Rebuttal · Authors · 2026-03-30
>
> We thank the reviewer for the insightful feedback. We address your specific concerns below:
>
> ### 1. Innovation & Equal-Budget Comparison (W1 & Q1)
> **Temporal Modeling.**  Unlike works focusing explicitly on sequential temporal modeling (e.g., Video-R1), ```our primary focus is on the paradigm shift in reasoning```—specifically, transitioning from naive empiricism to Bayesian inference. DUS and PTS are specifically designed to guide the model to execute this Bayesian inference during reinforcement learning. This shift in the reasoning paradigm and the corresponding guidance mechanism constitute our core innovation, setting our work apart from existing approaches. To summarize, our main contributions and advantages include (as outlined in the final paragraph of the Introduction):
> - Reasoning paradigm shift (Naive Empiricist -> Bayesian Inference);
> - Guidance design for Bayesian inference (DUS/PTS);
> - SOTA performance across all benchmarks (outperforming temporal modeling and consistency-based methods like Video-R1 and GRPO-CARE);
> - No requirement for extra manual annotations;
>
> **Equal K=8 Budget.** We evaluated a more powerful 32B baseline model on the MMVU. It achieves 62.0% with CoT and
> 67.2% under K=8 Majority Vote (self-consistency) In contrast, our 7B model achieves 68.0%, outperforming the 32B SC@8 (+0.8%). This proves RL guided by DUS/PTS fundamentally raises the reasoning upper bound beyond inference-time resampling (Reranking/SC). For the 7B baseline, see Qwen 2.5-VL-MJ in Table 2.
>
> ### 2. Source of Performance Gains (W2)
> ```the performance gain is rooted in the transformation of the reasoning (cognitive) paradigm```. A common failure mode in video understanding is that models easily fall into ```single-frame bias```. During training, our DUS mechanism dynamically penalizes this blind conformity, forcing the model to trace back through the complete video evidence to find cross-frame causal clues. Simultaneously, PTS solidifies these long-temporal causal reasoning chains. The significant gains we achieved on benchmarks emphasizing temporal and physical causality (e.g., MMVU and VideoMMMU) prove that ```reshaping the Bayesian inference paradigm``` activates the underlying model's temporal logical analysis potential, which was previously suppressed by single-frame bias.
>
> ### 3. Consistency of PTS Gains (W3 & Q3)
> In fact, the performance gains from PTS are stable. As shown in our ablation study (Table 2), both DUS and PTS effectively facilitate the RL process. Compared to the standard RL baseline (Qwen 2.5-BCI-GRPO), ```the gains introduced by PTS are substantial and consistent across all benchmarks```. For instance, on MMVU, Qwen 2.5-BCI-PTS outperforms Qwen 2.5-BCI-GRPO by +3.4%. The inconsistency you mentioned might be relative to DUS, which is also our proposed guidance method. Furthermore, on benchmarks that heavily require complex reasoning, Qwen 2.5-BCI-PTS shows a larger advantage over DUS. When DUS and PTS are used jointly to form our final model, they achieve a strong balance between reasoning-heavy and general-purpose benchmarks, while producing high-quality reasoning traces (we provide visualization analysis for all benchmarks in Appendix).
>
> ### 4. Bayesian Framework vs. Heuristic Design (W4)
> Nevertheless, we wish to clarify that framing this work within a Bayesian context is not merely a heuristic design, but rather ```the mathematical foundation for deriving our objective functions.``` Furthermore, computing a strict Bayesian posterior distribution is intractable for massive VLMs. Therefore, we construct an empirical approximation of the prior distribution through multi-hypothesis sampling. The designs of DUS and PTS are not heuristic rules; they strictly correspond to evaluating the information entropy (Cognitive Conflict) of the prior distribution and maximizing the alignment between the posterior and the evidence. Here, RL acts as the optimizer to realize the Bayesian integration. We will add a Discussion section in Section 3 to transparently discuss the trade-off between strict Bayesian updates and LLM-based RL approximations.
>
> ### 5. Experimental Details and Reproducibility (W5 & Q2)
> ```Regarding the similarity function``` PTS uses sentence-level semantic similarity (ROUGE-L F1). ```Sampling```: We avoided fine-grained tuning to highlight Video-BCI's intrinsic benefits, using standard RL rollout settings (T=1.0, top_p=0.95) aligned with Video-R1. To enhance reproducibility, ```we will open-source all designs, including parameter configurations and model weights```.
>
> These are our responses to your questions. We thank you again for your time and effort in reviewing our paper. If you have any further questions, please feel free to let us know.
>
> Sincerely,
>
> The Authors

---

> > ### Author Rebuttal · Reviewer_Ka56 · 2026-04-03
> >
> > The authors resolves my concerns, therefore i will raise my score

---

> > > ### Author Response · Authors · 2026-04-06
> > >
> > > We sincerely thank the reviewer for participating in the peer-review process and for conducting a thorough and rigorous evaluation of our manuscript. We greatly appreciate the scholarly expertise and meticulous dedication demonstrated throughout your assessment. Accordingly, we will carefully address each of your comments and incorporate the suggested revisions, ensuring that the final version presents more robust and reproducible findings.
> > >
> > > Once again, we are deeply grateful for your valuable contribution to the scholarly community and for your constructive engagement with our work.
> > >
> > > Sincerely,
> > >
> > > The Authors

---

### Decision · Program_Chairs · 2026-04-30

**Decision:**

Accept (regular)

**Comment:**

All reviewers gave positive comments, so I recommend acceptance.